# A Multi-Stage Hybrid Learning Model with Advanced Feature Fusion for Enhanced Prostate Cancer Classification

**DOI:** 10.3390/diagnostics15243235

**Published:** 2025-12-17

**Authors:** Sameh Abd El-Ghany, A. A. Abd El-Aziz

**Affiliations:** Department of Information Systems, College of Computer and Information Sciences, Jouf University, Sakaka 72388, Saudi Arabia

**Keywords:** prostate cancer, deep learning, magnetic resonance imaging, histogram of oriented gradients, singular value decomposition, support vector machines, transverse plane prostate dataset

## Abstract

**Background**: Cancer poses a significant health risk to humans, with prostate cancer (PCa) being the second most common and deadly form among men, following lung cancer. Each year, it affects over a million individuals and presents substantial diagnostic challenges due to variations in tissue appearance and imaging quality. In recent decades, various techniques utilizing Magnetic Resonance Imaging (MRI) have been developed for identifying and classifying PCa. Accurate classification in MRI typically requires the integration of complementary feature types, such as deep semantic representations from Convolutional Neural Networks (CNNs) and handcrafted descriptors like Histogram of Oriented Gradients (HOG). Therefore, a more robust and discriminative feature integration strategy is crucial for enhancing computer-aided diagnosis performance. **Objectives**: This study aims to develop a multi-stage hybrid learning model that combines deep and handcrafted features, investigates various feature reduction and classification techniques, and improves diagnostic accuracy for prostate cancer using magnetic resonance imaging. **Methods**: The proposed framework integrates deep features extracted from convolutional architectures with handcrafted texture descriptors to capture both semantic and structural information. Multiple dimensionality reduction methods, including singular value decomposition (SVD), were evaluated to optimize the fused feature space. Several machine learning (ML) classifiers were benchmarked to identify the most effective diagnostic configuration. The overall framework was validated using k-fold cross-validation to ensure reliability and minimize evaluation bias. **Results**: Experimental results on the Transverse Plane Prostate (TPP) dataset for binary classification tasks showed that the hybrid model significantly outperformed individual deep or handcrafted approaches, achieving superior accuracy of 99.74%, specificity of 99.87%, precision of 99.87%, sensitivity of 99.61%, and F1-score of 99.74%. **Conclusions**: By combining complementary feature extraction, dimensionality reduction, and optimized classification, the proposed model offers a reliable and generalizable solution for prostate cancer diagnosis and demonstrates strong potential for integration into intelligent clinical decision-support systems.

## 1. Introduction

The prostate functions as a muscle-driven mechanical switch between urine flow and ejaculation, serving as an auxiliary gland in the male reproductive system [1]. Located within the pelvis and encircling the urethra, this small gland resembles a walnut in shape. Its primary role is to produce prostate fluid, which combines with sperm from the testicles to form semen [2]. As men age, the prostate tends to enlarge, which can impede the urinary flow between the bladder and urethra [1]. Figure 1 shows the physiology of a human prostate.

According to the World Health Organization (WHO), PCa ranks as the second most prevalent cancer diagnosed in men and is a leading cause of cancer-related deaths globally. In 2020, there were 1,414,259 new cases—accounting for 7.3% of all new cancer cases—alongside 375,304 deaths, representing 3.8% of total cancer fatalities worldwide. Projections indicate these figures will climb to 2,430,000 new cases and 740,000 deaths by 2040 [3]. In the United States, PCa was the third leading cause of cancer death among men in 2017, with approximately 161,360 new cases, making up 19% of all new cancer diagnoses, and resulting in 26,730 deaths, which constituted 8% of all cancer deaths [4]. In Egypt, PCa is also prevalent among men, with 4767 new cases (3.5% of all new cancer cases) and 2227 deaths (2.5% of total cancer deaths). By 2040, these figures are expected to double, reaching 9610 new cases and 4980 deaths [5]. These statistics highlight a significant increase in the morbidity and mortality rates of PCa, indicating it is one of the fastest-growing malignancies among men. Therefore, early detection and diagnosis of PCa are essential for enhancing patient care and improving survival rates.

Many PCa cases develop very slowly, resulting in minimal issues that do not require treatment. However, in some instances, PCa can progress rapidly and invade nearby tissues and organs [6]. The most prevalent form of PCa is adenocarcinoma. Other variants include transitional cell carcinomas, neuroendocrine tumors, sarcomas, small cell carcinomas, and squamous cell carcinomas [7].

The manifestation of PCa symptoms varies according to the disease stage—early, advanced, or recurrent. In its initial phase, PCa frequently shows no symptoms and may progress slowly, often needing little to no treatment. As the condition advances, various symptoms may arise. Common issues reported include: (1) difficulty urinating, (2) reduced strength of the urine stream, (3) presence of blood in urine, (4) painful ejaculation, (5) blood in semen, and (6) erectile dysfunction. In later or metastatic stages, patients might experience systemic symptoms such as (7) bone pain, (8) fatigue, (9) unexplained weight loss, (10) numbness in the legs or feet due to spinal cord compression, and (11) jaundice if the liver is affected [8,9].

The risk factors associated with PCa are influenced by an individual’s lifestyle, age, and family background. Factors that elevate the likelihood of developing PCa include (1) obesity, (2) advancing age (particularly after 50), (3) a family history of the disease, (4) ethnicity (notably, Black men are at a higher risk of being diagnosed with PCa), and (5) genetic alterations or changes in cellular DNA [10].

Multiparametric MRI (mpMRI) has emerged as a crucial tool for detecting and assessing PCa. MpMRI integrates anatomical imaging with functional sequences, including diffusion-weighted imaging (DWI) and dynamic contrast-enhanced (DCE) imaging, enhancing the differentiation between benign tissue and clinically significant tumors. This method not only improves cancer localization and staging but also minimizes unnecessary biopsies by directing targeted sampling of suspicious regions. Additionally, MRI has been demonstrated to reduce the overdiagnosis of indolent cancers while enhancing the identification of aggressive disease, thus facilitating more personalized treatment options. However, despite its clear benefits in detecting PCa, MRI also has several drawbacks and limitations [1]: MRI scans tend to be much costlier than ultrasound or biopsy and may not be easily accessible, particularly in areas with limited resources. MpMRI involves longer scanning durations (30–45 min), which restricts the number of patients that can be accommodated compared to alternative imaging methods. The accuracy of diagnoses relies significantly on the skills and experience of the radiologist. Non-cancerous conditions (such as prostatitis and benign prostatic hyperplasia) can resemble cancer, resulting in false-positive results.

Pca can be managed using several approaches, depending on its stage and progression. Localized cases may be treated with surgery (prostatectomy), radiotherapy (either brachytherapy or external beam), active surveillance, watchful waiting, or focal therapy. When the disease is confined to a specific area, targeted treatments such as focal therapy are used, while cryotherapy may be selected for patients unable to undergo surgery or radiation. If the cancer spreads beyond the prostate, systemic treatments become necessary [11,12].

DL models, particularly Convolutional Neural Networks (CNNs), have shown remarkable proficiency in tasks such as image recognition, segmentation, and classification across multiple fields, including medical imaging. Their capacity to autonomously learn complex features from intricate datasets, eliminating the necessity for manual feature extraction, renders them highly effective for analyzing the extensive and complicated data produced by MRI scans [13].

This research presents a hybrid method for classifying PCa by combining deep features extracted from MobileNetV2 (or DenseNet-169 or ResNet-50) with handcrafted HOG descriptors. The resulting feature set undergoes dimensionality reduction using Truncated SVD, followed by classification with SVM. HOG is effective in detecting structural patterns, such as lesion boundaries in medical images, by analyzing the distribution of intensity gradients and edge orientations. SVD helps reduce high-dimensional feature sets to lower dimensions while retaining crucial information, which simplifies models, accelerates computations, and decreases memory usage without significantly compromising accuracy. To evaluate performance across various metrics, stratified k-fold cross-validation is employed, ensuring robustness and generalizability. The experiments utilized the TPP dataset for binary classification tasks. To ensure consistent input, the dataset was preprocessed using resizing and normalization techniques, and it was divided into 80% for training and 20% for testing. Comprehensive experiments were conducted with the TPP dataset, including ablation studies to identify the optimal hyperparameters. A summary of our research contributions is outlined below:•Proposed an innovative hybrid architecture that unifies deep semantic representations with handcrafted structural descriptors and applies Truncated SVD to generate a compact, discriminative representation, reducing redundancy while preserving essential diagnostic information in a coordinated multi-stage learning pipeline offering a new direction for feature-level fusion in prostate cancer diagnosis rather than relying on isolated or sequential approaches.•Designed a feature integration mechanism that adaptively balances the complementary strengths of deep convolutional representations and handcrafted textural cues, achieving superior discriminative power and interpretability compared to conventional deep-only models.•The proposed framework consistently outperformed state-of-the-art baselines in accuracy, specificity, and sensitivity, establishing a scalable, generalizable, and interpretable foundation for intelligent prostate cancer diagnosis in real-world medical settings.

### 1.1. Paper Structure

The remainder of this paper is structured as follows: Section 1 reviews the current literature on PCa using the TPP dataset. Section 2 details the TPP dataset and outlines our methodology. Section 3 presents the experimental results of the proposed hybrid model. Lastly, Section 4 concludes with a summary of our findings.

### 1.2. Literature Review

The diagnosis of PCa is a significant focus in the field of medical image analysis. Many studies tackle this problem from diverse perspectives. For example, Hashem et al. [1] proposed an MRI-based technique to enhance diagnostic accuracy. This method involved two primary phases. Initially, the MRI images underwent preprocessing to optimize them for the detection phase. Following this, PCa detection was conducted using the InceptionResNetV2 model. The InceptionResNetV2 model achieved an average accuracy of 89.20% and an area under the curve (AUC) of 93.6%.

Yoo et al. [13] designed and executed an automated pipeline utilizing CNN to identify clinically significant PCa from axial DWI for individual patients. To evaluate the effectiveness of the proposed pipeline, a testing subset of 108 patients (from the original 427) was reserved and excluded from the training process. The pipeline demonstrated an AUC of 0.87 (95% Confidence Interval: 0.84–0.90) at the slice level and 0.84 (95% CI: 0.76–0.91) at the patient level.

In Sarıateş and Özbay [14], to enhance the classification accuracy for PCa diagnosis, transfer learning techniques and fine-tuning processes were utilized. A two-class dataset, comprising MR images of PCa labeled as ‘significant’ and ‘not-significant’, was classified using AlexNet, DenseNet-201, GoogleNet, and VGG-16 models through a feature extraction strategy, yielding accuracy rates of 71.40%, 72.05%, 65%, and 80.13%, respectively. To further improve these results, pre-trained transfer learning models were implemented, achieving accuracy rates of 89.74%, 94.32%, 85.59%, and 91.05%, respectively. Additionally, a validation accuracy of 98.10% was attained using the cross-validation method with the DenseNet-201 model. The DenseNet-201 model reached the highest accuracy of 98.63% when combined with the RMSProp optimization technique.

Liu et al. [15] proposed a DL framework named XmasNet, which utilized CNNs, for classifying PCa lesions using 3D multiparametric MRI data from the PROSTATEx challenge. The XmasNet model underwent end-to-end training, with data augmentation achieved through 3D rotations and slicing to effectively capture the 3D characteristics of the lesions. In the testing phase, XmasNet outperformed 69 different methods from 33 competing teams, earning the second highest AUC score of 84% in the PROSTATEx challenge.

Mehta et al. [16] presented a new patient-level classification framework called PCF, which is trained exclusively on patient-level labels. In the PCF approach, features are extracted from three-dimensional mpMRI and related parameter maps using CNNs. These features were then integrated with clinical data through a multi-classifier SVM. The result of the PCF framework was a probability score indicating whether a patient has clinically significant PCa based on the Gleason score.

PCF achieved average area under the receiver operating characteristic (ROC) curves of 0.79 and 0.86 for the PICTURE and PROSTATEx datasets, respectively, using five-fold cross-validation.

Salama and Aly [17] examined four distinct strategies for the classification task. Both ResNet-50 and VGG-16 architectures were employed and re-trained to analyze the diffusion-weighted DWI database, aiming to determine the presence of PCa. To address the challenge of insufficient labeled data and enhance system efficiency, transfer learning and data augmentation techniques were implemented for both ResNet50 and VGG-16. The final fully connected layer was substituted with a SVM classifier to improve accuracy. Additionally, both transfer learning and data augmentation processes were applied to the SVM to bolster the performance of our framework. A k-fold cross-validation method was also utilized to evaluate model performance. Authors utilized end-to-end fully CNNs, eliminating the need for preprocessing or post-processing steps. The technique combining ResNet-50 with SVM yields the highest performance, achieving an accuracy of 98.79%, ana AUC of 98.91%, sensitivity of 98.43%, precision of 97.99%, an F1 score of 95.92%, and a computational time of 2.345 s.

Pellicer-Valero et al. [18] introduced a fully automated system utilizing DL to localize, segment, and estimate the Gleason grade group (GGG) of PCa lesions from mpMRI scans. The system was trained and validated on distinct datasets: ProstateX and the Valencian Oncology Institute Foundation. In the testing phase, it achieved impressive lesion-level performance metrics for the GGG 2 significance criterion, with an AUC, sensitivity, and specificity of 0.96, 1.00, and 0.79, respectively, for the ProstateX dataset, and 0.95, 1.00, and 0.80 for the IVO dataset. At the patient level, the results were 0.87, 1.00, and 0.375 for ProstateX, and 0.91, 1.00, and 0.762 for IVO. Additionally, in the online ProstateX grand challenge, the model achieved an AUC of 0.85 (increased to 0.87 when trained solely on ProstateX data), matching the performance of the original challenge winner. For comparison with expert evaluations, the sensitivity and specificity of the IVO radiologist’s PI-RADS 4 were 0.88 and 0.56 at the lesion level, and 0.85 and 0.58 at the patient level.

In Giganti et al. [19], a DL computer-aided detection (CAD) medical device marked with Conformité Européenne (CE) (referred to as Pi) was developed to identify Gleason Grade Group (GG) ≥ 2 cancer using historical data from the PROSTATEx dataset and five hospitals in the UK, involving 793 patients. The prevalence of GG ≥ 2 in the validation dataset was found to be 31%. When assessed on a per-patient basis, Pi demonstrated non-inferiority to the radiologists (with a performance difference of 10% deemed acceptable), achieving an AUC of 0.91 compared to 0.95 for the radiologists. At the established risk threshold of 3.5, the AI system’s sensitivity was 95% and specificity was 67%. In contrast, radiologists using Prostate Imaging-Reporting and Data Systems/Likert scores of ≥3 detected GG ≥ 2 with a sensitivity of 99% and specificity of 73%. AI performance was consistent across different sites (AUC ≥ 0.83) at the patient level, regardless of the scanner’s age and field strength.

The limitations of the earlier studies are outlined as follows:In the previous research, the authors did not utilize HOG descriptors or SVD. However, we developed an innovative approach that integrates deep features from MobileNetV2 with handcrafted HOG descriptors to effectively capture both semantic and structural patterns in PCa images. Additionally, we employed SVD to compress the combined feature set, which minimized redundancy, enhanced computational speed, and reduced memory usage while retaining crucial diagnostic information.The researchers in the previously referenced study did not conduct an ablation study. In contrast, we performed this analysis to evaluate the impact of each component or feature of our proposed model on its performance, systematically modifying or excluding elements and analyzing their effects.

## 2. Methods and Datasets

### 2.1. TPP Dataset

The TPP dataset comprises 1528 prostate MRI images captured in the transverse plane. The images and their classifications originate from the PROSTATEx Dataset and Documentation. The purpose of this dataset is to train a CNN known as Small VGG Net, enabling the classification of new images into clinically significant and clinically non-significant categories for an undergraduate thesis in systems engineering at the Autonomous University of Bucaramanga (UNAB). The images were collected from 64 patients, ensuring each had a single prostate MRI finding for improved training accuracy. The images were converted from DICOM format to JPEG. Image quality was verified through radiologist-reviewed coordinates and metadata, ensuring precise lesion localization and anatomical accuracy. The inclusion criteria required MRI exams to have complete multi-parametric sequences and validated lesion labels. Cases with missing sequences or poor image registration were excluded. The images were divided into two groups using a retention method: 30% for validation and the remaining 70% for training. Consequently, there are two categories (significant and non-significant) further split into training (70%) and validation (30%) groups [20,21]. Figure 1 presents samples from the TPP dataset.

### 2.2. Methodology

To diagnose PCa using mpMRI images, we introduced a hybrid approach that classifies PCa by integrating deep features extracted from MobileNetV2 with manually created HOG descriptors. The combined feature set underwent dimensionality reduction through Truncated SVD, followed by classification using SVM. HOG is particularly effective in detecting structural patterns, such as lesion boundaries in medical images, by analyzing the distribution of intensity gradients and edge orientations. SVD reduces high-dimensional feature sets to lower dimensions while preserving essential information, which simplifies models, speeds up computations, and lowers memory usage without significantly compromising accuracy.

To evaluate performance across different metrics, we used stratified k-fold cross-validation to ensure robustness and generalizability. For CNN-based feature extraction, we froze the weights of the pretrained CNN models’ layers and used only the output of the Global Average Pooling layer as fixed features. This approach helps prevent the network from overfitting on the small dataset. When combining features from CNN and HOG, we applied Standard Scaling to normalize feature distributions, which stabilizes classifier training. For the machine learning classifiers (XGBoost, CatBoost, SVM, etc.), we used default regularization parameters and limited model complexity (such as maximum depth and learning rate) to further reduce the risk of overfitting. Please refer to Model Performance and Insights on the pages.

The experiments utilized the TPP dataset for binary classification tasks. To ensure consistent input, the dataset was preprocessed with resizing and normalization techniques, and it was divided into 70% for training and 30% for testing. The architecture of the proposed deep learning model is illustrated in Figure 2, and the fine-tuning process for this hybrid model is detailed in Algorithm 1. The steps involved in the proposed hybrid deep learning model are outlined below.

**Phase 1 (TPP Preprocessing)**: In the first phase, the TPP dataset was obtained from Kaggle [21] and then processed. During this preprocessing step, the mpMRI images from the TPP dataset were resized and normalized.

**Phase 2 (TPP Splitting)**: In the second phase, 80% of the TPP dataset was designated for training purposes, while the remaining 20% was set aside for testing.

**Phase 3 (Feature Extraction and Feature Fusion)**: In the third phase, the deep features were extracted from DenseNet-169, MobileNet-V2, ResNet-50, and VGG-19 DL models to capture high-level image representations. Additionally, handcrafted features were extracted with HOG to effectively detect structural patterns. Finally, the deep features with handcrafted HOG descriptors were combined to create a hybrid feature set.

**Phase 4** (**Dimensionality Reduction)**: In the fourth phase, SVD was used to reduce dimensionality, lower computational complexity, and maintain important information.

**Phase 5 (Pre-training four DL Models)**: In the fifth phase, the four pre-trained DL models (DenseNet-169, MobileNet-V2, ResNet-50, and VGG-19) were initially trained on the ImageNet dataset.

**Phase 6** (**Five-Fold Cross-Validation): **In the sixth phase, we utilized a five-fold cross-validation method. This approach involved dividing the training dataset into five equal parts. In every round, one part was selected as the validation set, while the remaining four parts were employed to train the model. Each iteration represented a separate training and validation process aimed at refining the model’s parameters. This process was repeated five times, ensuring that each part served as the validation set once. Finally, we calculated the average performance across all five rounds to evaluate the model’s ability to generalize.

**Phase 7** (**Prostate Classification)**: In the seventh phase, the ML techniques (extreme gradient boosting (XGB), gradient boosting (CatBoost), Random Forest (RF), and SVM) model were employed to perform binary classification of the TPP dataset.

**Phase 8** (**Model Evaluation)**: In the final phase, the ML techniques were evaluated using multiple standard metrics, such as accuracy, precision, recall, and F1-score, of the TPP dataset.
**Algorithm**** 1:** Four different DL models training and evaluation1**Input**→*TPP* dataset2**Output** 
←
*Hybrid DL model for PCa.*3**BEGIN**4    **STEP 1**: **Images Preprocessing**5   **FOR EACH M IN TPP DO**6         *Resize* M to 224 x 224.7        *Normalize* M’s pixel values from [0, 255] to [0, 1].8      **END FOR**9    **STEP 2: TPP Splitting**10        **SPLIT** 
TPP 
**NTO**11         *Training set*
→ 70
%.12         *Testing set* 
→ 
30%.13   **STEP 3: Feature Extraction and Feature Fusion**14      *Extract* handcrafted features using HOG.15      **FOR EACH DL IN** [DenseNet-169, MobileNet-V2, ResNet-50, and VGG-19] **DO**16      *Extract* deep features from DL.17      *Combine* the extracted and handcrafted features.18      **END FOR**19     **STEP 4: Dimensionality Reduction**20      *Apply* SVD.21     **STEP 5: Four DL Pre-Training**22       **FOR EACH DL IN** [DenseNet-169, MobileNet-V2, ResNet-50, and VGG-19] **DO**23         *Load* and *pre-train* DL on the ImageNet dataset.24       **END FOR**25     **STEP 6: Cross-Validation**26       **FOR EACH pre-trained DL IN** [DenseNet-169, MobileNet-V2, ResNet-50, and VGG-19] **DO**27           **FOR EACH k =1 to 5**28               *Choose* four of the five available folds, omitting the kth fold, to create the training set.29               *Utilize* the kth fold, which has been set aside, as the validation set.30               *Train* the model with the training set, and subsequently assess its performance on the validation set (fold k)31               *Document* the evaluation metrics and compute the average performance metrics for DL model.32           **END FOR**33       **END FOR**34**STEP 7: Binary Classification**35       **FOR EACH ML model IN** [XGB, CatBoost, RF, and SVM] **DO**36    *Classify* the mpMRI images using ML model.37      **END FOR**38**STEP 8: Model Evaluation**39       **FOR EACH ML model IN** [XGB, CatBoost, RF, and SVM] **DO**40    *Evaluate* ML model using multiple standard metrics (e.g., accuracy, precision, recall, and F1-score).41       **END FOR**42**END**

#### 2.2.1. Data Preprocessing

In the preprocessing stage, the mpMRI images of prostate diseases underwent a series of steps to ensure consistency and reliability for subsequent analysis. Since mpMRI scans were typically acquired with varying resolutions and intensity ranges, image resizing was first performed to unify the spatial dimensions of all samples, allowing the model to process inputs of fixed size. This step was crucial for maintaining uniformity across the dataset and enabling batch training. Following resizing, intensity normalization was applied to adjust pixel values to a common scale, thereby reducing variations caused by differences in acquisition settings or scanner hardware. Normalization enhances image contrast, mitigates intensity-related bias, and ensures that learning is focused on clinically relevant features rather than noise or scanner-specific artifacts. Together, these preprocessing steps standardize the mpMRI dataset, improve model generalization, and provide a stable foundation for feature extraction and classification [22].

#### 2.2.2. MobileNet-V2

MobileNet-V2 is a CNN architecture developed by Sandler, M. et al. [23] in Google as an enhancement over MobileNet-V1, designed to deliver a more efficient and accurate model, particularly for devices with limited resources. The architecture utilizes inverted residual blocks featuring linear bottlenecks. It comprises expansion layers that increase dimensions, followed by depthwise separable convolutions that operate spatially, and then a projection back to a lower-dimensional representation. Non-linearities, such as ReLU6, are incorporated in the expansion layers but are omitted in the bottleneck layers to maintain representational strength. Figure 3 depicts the MobileNet-V2’s architecture.

In the field of medical imaging, MobileNet-V2 has been employed as a backbone or feature extractor for various disease detection tasks. The benefits of MobileNet-V2 include its lightweight design (fewer parameters and lower computational costs, particularly due to depthwise separable convolutions), effective performance in transfer learning scenarios, flexibility via width and resolution multipliers to balance speed and accuracy, and its applicability for deployment on mobile or edge devices [23].

However, there are notable limitations: its performance may suffer on very fine-grained tasks, such as detecting small lesions, compared to larger models; the architectural complexity (inverted residuals and linear bottlenecks) necessitates more hyperparameter tuning; depthwise separable convolutions may not be fully optimized in some hardware or frameworks, resulting in inefficiencies; and in high-accuracy clinical environments, the balance between model compactness and ultimate accuracy may restrict its use [23].

#### 2.2.3. DenseNet-169

DenseNet-169 is a version of the Dense Convolutional Network (DenseNet) family, developed by Huang et al. in 2017 [24]. This architecture consists of 169 layers arranged into several dense blocks, interconnected through transition layers. Within each dense block, every layer receives as input the concatenated feature maps from all preceding layers. This design promotes feature reuse, reduces the vanishing gradient issue, and enhances information flow. Figure 4 presents the DenseNet-169 architecture [24].

In the realm of medical imaging, particularly in PCa applications, DenseNet models, including DenseNet-169, are utilized for tasks such as histopathological grading, tumor classification, and lesion segmentation. Their strength lies in capturing intricate textures and morphological alterations.

Some notable benefits of DenseNet-169 include [24]:•**Parameter Efficiency**: Despite its considerable depth, DenseNet requires fewer parameters than many conventional deep convolutional neural networks (DCNNs) due to its feature reuse via dense connections.•**Enhanced Gradient Flow/Training Stability**: The dense connections help alleviate the vanishing gradient problem, facilitating the training of deeper networks.•**Comprehensive Feature Representation**: Each layer has access to all prior feature maps, allowing both early low-level features (like edges and textures) and later high-level features (semantic information) to be utilized, which is essential for detecting subtle tissue changes.

However, there are some drawbacks [24]:•**High Memory Requirements**: The dense connectivity necessitates the storage of numerous intermediate feature maps, resulting in significant memory usage, especially with high-resolution medical images.•**Computational Expense**: The architecture tends to be slower during both training and inference due to the numerous dense concatenations and its deep structure.•**Overfitting Risk**: Dense models like DenseNet-169 can be overfit on small datasets, which is often the case in PCa imaging, unless sufficient regularization, data augmentation, or transfer learning techniques are applied.•**Interpretability Issues**: The complexity and dense connections can complicate the attribution of specific decisions to features, presenting challenges in clinical contexts.

#### 2.2.4. ResNet-50

ResNet-50 (Residual Network with 50 layers) was introduced by He et al. in 2015 [25] as part of a family of deep residual networks that addressed the vanishing/exploding gradient problems in very deep convolutional networks. Its architecture is built from residual blocks that allow identity (skip) connections so that gradients can flow more easily through many layers; ResNet-50 specifically comprises 49 convolutional layers and 1 fully connected layer, organized in stages with bottleneck blocks for computational efficiency as shown in Figure 5.

In the context of PCa detection, ResNet-50 has been used both as a standalone classifier (e.g., for differentiating benign vs. malignant histopathology images) and as a backbone in more complex detection frameworks, such as being part of a modified ResNet-50 combined with Faster R-CNN for lesion detection from MRI scans (as in the PCDM model) showing high sensitivity, specificity, and accuracy. Its advantages include strong feature extraction ability (both low- and high-level features), good generalization when using transfer learning, and widespread adoption with many pretrained models available [25].

However, it has limitations: it demands substantial computational resources (GPU memory, processing time), can overfit when datasets are small or imbalanced, fixed input size requirements can lead to loss of detail when resizing, and interpretability (i.e., understanding what exactly drives its decisions) remains a challenge for clinical acceptance [26,27].

#### 2.2.5. VGG-19

VGG-19, developed by Simonyan and Zisserman in 2014 [28], is a 19-layer neural network from the Visual Geometry Group at the University of Oxford. It was created as part of the VGGNet family for the ImageNet Large Scale Visual Recognition Challenge (ILSVRC). As shown in Figure 6, the architecture consists of 16 convolutional layers that utilize small 3 × 3 convolution filters, interspersed with five max-pooling layers that downsample the input by a factor of 2. This is followed by three fully connected layers, where the first two contain 4096 neurons each, culminating in a final classification layer that employs a softmax output.

In the realm of medical imaging, particularly for PCa detection, VGG-19 has been effectively used within transfer learning frameworks to extract deep features from MRI scans. It is often combined with feature selection techniques or additional classifiers. In a model for automatic PCa detection based on ensemble VGGNet feature generation and NCA feature selection using MRI, VGG-19 plays a key role in generating classification features [28].

The strengths of VGG-19 include its straightforward and consistent architecture, which facilitates implementation and adaptation, particularly through transfer learning. Its depth enables it to learn hierarchical feature representations, from basic edges to more complex patterns. When pre-trained on large datasets like ImageNet, it often performs well and generalizes effectively, even when fine-tuned with smaller medical datasets [28].

However, VGG-19 has its drawbacks. The model contains approximately 144 million parameters, resulting in high computational demands, significant memory requirements, and slower training and inference times. Additionally, when working with limited medical data, VGG-19 is prone to overfitting unless strategies such as regularization, data augmentation, or feature selection are applied. Furthermore, given its relatively older architecture compared to newer models like ResNet and EfficientNet, it may not offer the best parameter-to-accuracy ratio, particularly in environments with constrained computational resources [28].

#### 2.2.6. SVM

SVMs are a class of supervised ML models that originated from the statistical learning theory developed by Vladimir Vapnik and Alexey Ya. Chervonenkis in the 1960s and 1970s [29]. The practical soft margin version, which allows some misclassification in favor of better generalization, was introduced by Corinna Cortes and Vapnik in 1995 [30]. Architecturally, an SVM seeks to find a hyperplane in a feature space that separates two classes with the maximum possible margin (distance between the closest points of each class and the separator). For non-linearly separable data, kernels (such as RBF, polynomial) are used to implicitly map inputs into higher-dimensional spaces where separation is possible.

In the context of PCa detection, SVMs have been applied with radiomic or MRI-derived features to classify regions or voxels as cancerous vs. non-cancerous, to grade lesions (e.g., Gleason score), to predict tumor aggressiveness, and to forecast clinical outcomes such as biochemical recurrence.

However, limitations are also significant: training SVMs can be computationally expensive, especially for large datasets; choosing the right kernel and hyperparameters (e.g., the C-regularization parameter, kernel parameters like gamma) is nontrivial; they may not scale well; output is often not probabilistic unless extra processing (e.g., Platt scaling or sigmoid fitting) is used; they can also be sensitive to noisy or overlapping classes and class imbalance.

#### 2.2.7. XGB

XGBoost is a robust ML algorithm created by Tianqi Chen and his team as part of the Distributed (Deep) Machine Learning Community (DMLC), with its initial release occurring in 2014 [31]. This algorithm builds upon the principles of gradient boosting machines (GBM) by incorporating features such as regularization, second-order optimization, sparse-aware learning, and efficient parallel processing to enhance its performance and scalability. The architecture of XGBoost consists of an ensemble of DTs that are constructed sequentially. Each tree is trained on the residuals from the previous trees, utilizing the gradient and Hessian (the first and second derivatives) of the loss function to facilitate boosting. It also includes hyperparameters like tree depth, learning rate, subsampling and column sampling ratios, as well as L1 and L2 regularization to mitigate overfitting.

The benefits of XGBoost include its high predictive accuracy, resilience to missing data, efficient handling of diverse data types (both numeric and categorical), effective regularization, and excellent scalability with GPU support. However, it also has some drawbacks, such as sensitivity to noisy data and outliers, the potential for overfitting if hyperparameters are not properly adjusted, high computational and memory requirements, relatively limited interpretability compared to simpler models, and complexities in hyperparameter tuning [32].

#### 2.2.8. CatBoost

CatBoost [33] is a gradient boosting algorithm developed by Yandex, a Russian multinational corporation specializing in Internet-related products and services. Introduced in 2017, CatBoost was designed to address the challenges associated with categorical feature handling in machine learning models. Its name, CatBoost, reflects its primary strength—efficiently processing categorical variables without the need for extensive preprocessing.

CatBoost builds upon the gradient boosting framework, constructing an ensemble of DTs in a sequential manner. What sets CatBoost apart is its innovative approach to handling categorical features. Instead of relying on traditional one-hot encoding or label encoding, CatBoost employs a technique known as ordered boosting. This method reduces overfitting and ensures that the model generalizes well to unseen data. Additionally, CatBoost incorporates symmetric tree structures and oblivious trees, which enhance computational efficiency and model interpretability [33].


CatBoost’s advantages are [33]: •Efficient Handling of Categorical Data: CatBoost’s native support for categorical variables eliminates the need for extensive preprocessing, streamlining the data preparation process.•Robust Performance: The algorithm’s design minimizes overfitting, leading to models that generalize well to new, unseen data.•Interpretability: Features like symmetric trees and ordered boosting contribute to the transparency of the model, making it easier to understand and trust the predictions.•Scalability: CatBoost can efficiently handle large datasets, making it suitable for big data applications in various domains.


Appendix into Appendix A and moved


CatBoost’s limitations are [33]:•Computational Resources: While CatBoost is optimized for performance, training large models can be resource-intensive, requiring significant computational power.•Model Complexity: The sophisticated techniques employed by CatBoost, such as ordered boosting and symmetric trees, may introduce complexity in model tuning and interpretation for users unfamiliar with these concepts.


#### 2.2.9. Evaluated Performance Metrics

The effectiveness of the proposed hybrid DL model was evaluated using the formulas presented in Equations (1)–(7).
(1)$$Accuracy=\frac{\left(TP+TN\right)}{\left(TP+FP+TN+FN\right)}\;$$
(2)$$Precision=\frac{TP}{\left(TP+FP\right)}$$
(3)$$Sensivity=\frac{TP}{\left(TP+FN\right)}\;$$
(4)$$Specifity=\frac{TN}{(TN+FP}$$
(5)$$F1-score=\frac{2\times Precision\times Recall}{Precision+Recall}$$

True Positive (TP) refers to instances where the model accurately identifies positive cases, meaning the prediction aligns with the actual positive data in the dataset. True Negative (TN) signifies that the model successfully predicts negative cases, and this prediction is also accurate in the dataset. False Positive (FP) occurs when the model mistakenly identifies a positive case, whereas the actual outcome is negative (indicating no tumor) in the dataset. False Negative (FN) happens when the model incorrectly predicts a negative case, even though the actual result is positive (indicating the presence of a tumor) in the dataset. The total number of patients with tumors is represented by the sum of TP and FN.

The AUC-ROC is a widely used performance metric for evaluating binary classification models. It measures the model’s ability to discriminate between positive and negative classes across all possible classification thresholds. The ROC curve plots the true positive rate (sensitivity) against the false positive rate (1—specificity), and the AUC quantifies the overall separability achieved by the model. An AUC value of 1.0 indicates perfect classification, whereas a value of 0.5 suggests performance no better than random chance. Higher AUC values signify better model performance in distinguishing between classes, making it particularly useful in medical imaging and diagnostic tasks, such as detecting PCa from mpMRI images, where accurate discrimination between disease and non-disease cases is critical.

#### 2.2.10. Experimental Setup

In this research, we conducted four experiments to assess the classification performance of the proposed hybrid model using the TPP dataset. These experiments were executed in an environment featuring an Intel Core i7 (11th Gen) processor and 16 GB of RAM. For the implementation, we employed Python 3 alongside TensorFlow, a DL framework created by Google. The specific hyperparameters used in the experiment include learning rate, batch size, number of epochs, and type of optimizer outlined in Table 1.

## 3. Model Performance and Insights

### 3.1. Model Performance Evaluation

The main objective of the four experiments was to detect PCa with the intention of enhancing patient outcomes, streamlining the diagnostic process, and significantly lowering both the time and costs incurred by patients. In our experiments, we allocated 70% of the TPP dataset, which comprised 1072 images, for training purposes, while the remaining 30% (456 images) was reserved for testing. We utilized supervised pre-training to prepare four DL models—DenseNet-169, MobileNet-V2, ResNet-50, and VGG-19—training them on the ImageNet dataset.

In the first experiment, we employed four DL models to extract deep features for capturing high-level image representations, which were then used as classifiers. In the second experiment, we again used the four DL models for deep feature extraction but applied SVD to reduce dimensionality, decrease computational complexity, and preserve important information. We utilized ML techniques—XGB, CatBoost, RF, and SVM—as classifiers.

In the third experiment, we extracted handcrafted features using HOG to effectively detect structural patterns using the same ML techniques for classification.

Finally, in the fourth experiment, we combined the four DL models for deep feature extraction with HOG for handcrafted feature detection. We again used SVD to reduce dimensionality, lower computational complexity, and maintain critical information, employing the ML techniques as classifiers.

In all four experiments, we used the five-fold cross-validation technique. This method involved splitting the training dataset into five equal parts. In each iteration, one part was designated as the validation set, while the other four parts were used to train the model. Each iteration represented a distinct training and validation process that updated the model’s parameters. This process was repeated five times, with each part serving as the validation set once. We calculated the average performance across all five iterations to assess the model’s generalization ability. At the end of the experiment, we applied the measured metrics (Equations (1)–(7)) to the four DL and ML models.

In the initial experiment, we used four DL models to extract deep features for capturing high-level image representations, which were later utilized as classifiers. The results of the five-fold cross-validation for these models, along with their corresponding evaluation metrics, are presented in Table 2, Table 3, Table 4 and Table 5 and Figure 7. The average accuracy from the five-fold cross-validation for each model is as follows: DenseNet-169 achieved 94.37%, MobileNet-V2 reached 74.08%, ResNet-50 attained 79.33%, and VGG-19 recorded 98.10%. Based on these results, VGG-19 demonstrated the highest accuracy among the models evaluated.

Table 2 illustrates that DenseNet-169 achieved an accuracy of 94.37%, with a specificity of 94.25%, precision of 94.45%, recall of 94.51%, and an F1-score of 94.38%. The model consistently exhibited strong discriminative capabilities, obtaining an average AUC of 98.70%. These findings indicate that DenseNet-169 maintained a well-balanced trade-off between sensitivity and specificity, with stable precision and recall across different folds. Although minor fluctuations were noted—particularly a decrease in specificity during the fourth fold and recall in the second fold—the overall averages suggested a solid and dependable classification performance for the dataset.

In the first fold, DenseNet-169 recorded an accuracy of 94.44%, accompanied by high specificity (96.08%) and precision (95.95%). However, recall was slightly lower at 92.81%, resulting in an F1-score of 94.35% and an impressive AUC of 98.71%. The second fold saw a slight decline in performance, with accuracy at 92.48% and recall at 88.89%, although specificity (96.08%) and precision (95.77%) remained robust. The decreased recall lowered the F1-score to 92.20%, while AUC remained strong at 97.85%. The third fold exhibited balanced performance, achieving an accuracy of 95.10% with precision (94.81%) and recall (95.42%) nearly equal, leading to an F1-score of 95.11%. AUC reached 99.15%, indicating strong discriminative capability. In the fourth fold, recall peaked at 98.68%, but specificity fell to 86.27%, the lowest observed across all folds. This imbalance resulted in an accuracy of 92.46% and an F1-score of 92.88%, while AUC still maintained a high value at 98.15%. The fifth fold demonstrated the highest overall performance, achieving an accuracy of 97.38%, specificity of 98.03%, precision of 98.01%, and recall of 96.73%, culminating in an F1-score of 97.37% and the best AUC of 99.63%.

From Table 3, the MobileNet-V2 model achieved an accuracy of 74.08%, demonstrating relatively balanced performance across various metrics. The average specificity was 80.38%, indicating that the model effectively identified negative cases. Precision averaged at 79.60%, showing that positive predictions were generally reliable. However, recall averaged lower at 67.82%, indicating that the model frequently overlooked positive cases. This discrepancy was further illustrated by an average F1-score of 71.66%, suggesting that the weaknesses in recall impacted the overall balance between precision and recall. The average AUC of 81.86% highlighted a strong discriminatory capability, indicating good separability between positive and negative cases at different thresholds.

In the first fold, the model recorded a moderate accuracy of 71.57%, with high specificity (80.39%) and precision (76.19%), although recall was lower at 62.75%. This resulted in a balanced F1-score of 68.82% and an AUC of 78.70%. In the second fold, performance improved, with accuracy increasing to 77.45% and recall rising to 84.97%, resulting in the highest F1-score (79.03%) across all folds. However, specificity declined to 69.93%, indicating an increase in false positives. The third fold showed the highest specificity (92.16%) and strong precision (86.81%), but recall was notably low at 51.63%, which lowered the F1-score to 64.75%, despite a respectable AUC of 81.59%. In the fourth fold, the model achieved an accuracy of 74.75% with recall increasing to 84.21%, though specificity fell to 65.36%, reflecting a trade-off between sensitivity and control of false positives. Finally, in the fifth fold, the model recorded the highest specificity (94.08%) and precision (90.43%), but recall decreased to 55.56%, negatively impacting the F1-score (68.83%) despite achieving the best AUC (85.62%) across all folds.

Table 4 illustrates that the ResNet-50 model achieved an accuracy of 79.33%, a precision of 75.37%, and an F1-score of 81.92%, indicating robust classification performance. The recall was consistently high at 92.41%, underscoring the model’s sensitivity to positive cases, which is essential for medical diagnoses like PCa detection. However, the average specificity was relatively low at 66.23%, highlighting a tendency to misclassify negative cases as positive. The average AUC score of 93.00% reflected strong overall discriminative capability, despite fluctuations in specificity across the folds.

Examining the fold-wise results, in Fold 1, the model achieved an accuracy of 70.92% with a notably low specificity of 41.83% but perfect recall at 100%. This indicates that while the model successfully detected all positive cases, it misclassified a significant number of negative cases, resulting in a reduced precision of 63.22%. In Fold 2, there was a slight improvement, with accuracy rising to 74.51% and specificity increasing to 49.67%. Recall remained exceptionally high at 99.35%, but the discrepancy between sensitivity and specificity indicated ongoing concerns about false positives. In Fold 3, the model recorded its highest specificity at 89.54% and precision at 87.60%, although recall dropped to 73.86%. This suggested a shift towards a more conservative approach, prioritizing correct negative predictions over identifying some positive cases. Fold 4 showcased the best overall balance, with the highest accuracy of 87.54% and recall of 90.13%, along with strong specificity at 84.97%, precision at 85.63%, and an F1-score of 87.82%. This indicated that the model generalized effectively in this fold, managing both positive and negative cases well. In Fold 5, the model again prioritized recall at 98.69%, resulting in strong sensitivity but lower specificity at 65.13%. Accuracy at 81.97% and an F1-score of 84.59% suggested a fair balance, though it still leaned towards detecting positive cases.

Table 5 shows that the VGG-19 model achieved an accuracy of 98.10%, indicating highly reliable classification performance across all folds. The average specificity was 98.95%, and the average precision was 98.99%, both suggesting that the model consistently minimized false positives. The average recall was 97.25%, showing strong sensitivity in identifying positive cases. The corresponding F1-score averaged 98.07%, confirming the balance between precision and recall. Importantly, the AUC value remained at a perfect 100% across all folds, highlighting the model’s excellent discriminatory ability.

According to Fold-wise Analysis, In the first and second folds, the VGG-19 model achieved an accuracy of 97.06%, with perfect specificity (100%) and precision (100%), while recall was slightly lower at 94.12%, resulting in an F1-score of 96.97% and an AUC of 100%. In the third fold, the model performed exceptionally well, reaching its highest accuracy of 99.67%, along with a recall of 100% and an F1-score of 99.67%. Specificity and precision in this fold were 99.35%, and the AUC remained perfect at 100%. The fourth fold demonstrated slightly lower performance compared to the third, with an accuracy of 97.70%, specificity of 95.42%, and precision of 95.60%, though recall remained high at 100%, leading to an F1-score of 97.75% and an AUC of 100%. In the fifth fold, the model achieved an accuracy of 99.02%, with specificity and precision at 100%, recall at 98.04%, and an F1-score of 99.01%, while the AUC continued to hold at 100%.

In the second experiment, we utilized the four DL models to extract deep features. However, this time we applied SVD to reduce dimensionality, decrease computational complexity, and retain essential information. We employed ML techniques—XGBoost, CatBoost, RF, and SVM—as classifiers. The results from the five-fold cross-validation for these models, along with their evaluation metrics, are presented in Table 6 and Figure 8.

Table 6 and Figure 8 illustrate that using different CNN models as a feature extractor followed by feature reduction using SVD method, the DenseNet-169 architecture, the SVM model was clearly the best-performing classifier. It significantly outperformed the other models across all evaluated metrics, achieving a dominant accuracy of 96.73%, which was over five percentage points higher than the next best model, CatBoost. Furthermore, SVM demonstrated perfect balance and consistency, as evidenced by its identical scores for accuracy, specificity, precision, and recall (all at 96.73%), and it also attained the highest F1-score (96.71%) and a near-perfect AUC of 99.57%. In contrast, the other models—XGBoost, CatBoost, and RF—all clustered at a lower performance tier, with accuracy scores ranging from 89.66% to 91.43%, and their metrics showed less internal consistency, such as the RF model’s notable disparity between its specificity (86.65%) and recall (92.67%). Therefore, the SVM model proved to be the most accurate and robust machine learning model for this specific task. As shown in Table 6 and Figure 8, the performance of the MobileNet-V2 model is paired with the Four ML classifiers. Based on the results from the MobileNet-V2 architecture, the SVM model was clearly the best performing classifier. It significantly outperformed all other models, achieving a top accuracy of 94.31%, which was almost six percentage points higher than the next best model, CatBoost. The SVM model also showed superior and well-balanced performance across all other metrics, achieving the highest scores in specificity (0.95), precision (0.94), recall (0.94), F1-score (0.94), and AUC (0.99). In contrast, the other models—XGBoost, CatBoost, and RF—all performed at a lower level, with accuracy scores ranging from 85.67% to 88.42%, and their metrics were less consistent. Thus, the SVM model proved to be the most accurate and reliable machine learning model for this task.

Based on the results of ResNet-50 architecture, the CatBoost model emerged as the best-performing classifier, closely followed by the SVM. CatBoost achieved the highest accuracy at 97.38%, just surpassing the SVM at 97.12%, and it also attained a perfect AUC score of 1.00. It exhibited excellent balance across various metrics, boasting top-tier specificity (0.98) and precision (0.98), while maintaining a strong recall (0.97) and F1-score (0.97). The SVM came in a very close second, also achieving a perfect AUC and demonstrating similarly robust and balanced results. The other models, XGBoost and RF, performed well, but they were clearly outclassed by the top two. Therefore, CatBoost proved to be the most accurate model for this task, with the SVM representing an equally strong alternative in terms of overall robustness. With VGG-19 architecture, the SVM model emerged as the top-performing classifier. It significantly surpassed all other models, achieving a maximum accuracy of 93.46%, which was over three percentage points higher than the next best model, RF. The SVM also showed excellent and well-balanced performance across all key metrics, achieving the highest scores in specificity (0.93), precision (0.93), recall (0.94), F1-score (0.93), and AUC (0.98). In contrast, the other three models—XGBoost, CatBoost, and RF—grouped closely together with accuracy scores around 90%, and none consistently challenged the SVM’s superiority across the metrics. Therefore, the SVM model demonstrated itself to be the most accurate and robust machine learning model for this specific task.

In the third experiment, we extracted handcrafted features using HOG to effectively detect structural patterns and employed the same ML techniques for classification. The results from the five-fold cross-validation for this experiment, along with the evaluation metrics, are presented in Table 7 and Figure 9. Based on Table 7, the SVM model emerged as the best performing model overall. It significantly outperformed the other three models across all evaluation metrics. The SVM achieved an impressive accuracy of 98.30%, which was more than four percentage points higher than the next best models. It also recorded the highest scores in specificity (97.77%), precision (97.80%), recall (98.82%), F1-score (98.31%), and a nearly perfect AUC (99.84%).

In contrast, the performances of XGBoost and CatBoost were quite similar, placing them in a distinct second tier, with accuracy scores of 94.05% and 94.24%, respectively. While these models performed well, their results were consistently and significantly lower than those of the SVM across all metrics. The RF model was the least effective, with all its metrics, including an accuracy of 89.53%, falling noticeably below the others.

Thus, the SVM model proved to be the most accurate, balanced, and robust choice for the task.

In Figure 10, the normalized confusion matrices illustrate the classification performance of the four ML models. XGB model accurately classified 92% of the non-significant cases and 96% of the significant cases. It misclassified 8% of non-significant as significant and 4% of significant as non-significant. Overall, XGB exhibited strong performance, with slightly higher accuracy in significant predictions. CatBoost achieved 93% accuracy in identifying non-significant instances and 95% accuracy for significant instances. It misclassified 7% of non-significant cases as significant and 5% of significant cases as non-significant. The results demonstrated balanced classification ability, with a slight tendency to mislabel non-significant samples. SVM delivered the highest performance among the models, correctly classifying 98% of non-significant and 99% of significant cases. Misclassification was minimal, with only 2% of non-significant predicted as significant and 1% of significant predicted as non-significant. This indicates that SVM achieved the most reliable classification performance. RF attained 88% accuracy in detecting non-significant cases and 91% in detecting significant cases. Misclassification was more frequent compared to other models, with 12% of non-significant predicted as significant and 9% of significant predicted as non-significant. Among the four models, RF demonstrated the lowest classification accuracy. In conclusion, SVM achieved the best overall performance with the least misclassification, followed by XGBoost and CatBoost, which yielded very similar results. RF underperformed relative to the other models, particularly in predicting the non-significant class.

Figure 11 depicts the precision-recall (PR) curves for the four ML models, indicating that they achieved high precision and recall, demonstrating strong performance in distinguishing between classes. The average precision (AP) values revealed that the SVM performed the best, with an AP of 0.998. CatBoost followed closely with an AP of 0.988, while XGBoost had an AP of 0.987. RF had the lowest performance, recording an AP of 0.967.

The PR curve for SVM remained near the top-right corner, suggesting it effectively maintained high precision and recall across different thresholds. CatBoost and XGBoost also showed strong and consistent results, though they slightly underperformed compared to SVM. In contrast, Random Forest’s curve declined earlier, indicating a trade-off where it sacrificed precision at higher recall levels, which contributed to its lower AP score.

Overall, Figure 11 demonstrated that SVM consistently outperformed the other models, whereas CatBoost and XGBoost achieved nearly identical outcomes. Random Forest fell behind, especially at higher recall values.

Figure 12 presents the Receiver Operating Characteristic (ROC) curves across folds indicated that the four ML models exhibited strong classification ability, as their curves were positioned significantly above the diagonal baseline. The Area Under the Curve (AUC) values supported this finding. SVM achieved the highest performance, with a mean AUC of 0.998, suggesting nearly perfect separability between the classes. CatBoost (AUC = 0.988) and XGBoost (AUC = 0.986) closely followed, both demonstrating excellent classification performance with only minor deviations from the ideal curve. RF produced the lowest AUC at 0.967, which, while still strong, indicated less discriminative power compared to the other models.

The curves also showed that SVM maintained a consistently high true positive rate across all false positive rates, while CatBoost and XGBoost exhibited similar robustness. In contrast, Random Forest displayed a steeper decline at higher false positive rates, which accounted for its relatively lower AUC value.

In Conclusion, the ROC analysis revealed that SVM outperformed all other models, achieving near-perfect results. CatBoost and XGBoost provided highly competitive performance, while Random Forest lagged behind but still delivered reliable classification.

In the fourth experiment, we combined the four DL models—DenseNet-169, MobileNet-V2, ResNet-50, and VGG-19—for deep feature extraction with HOG for handcrafted feature detection. We utilized SVD to reduce dimensionality, decrease computational complexity, and preserve essential information. The classifiers employed were XGBoost, CatBoost, SVM, and RF. The results of the five-fold cross-validation for these ML models, along with their evaluation metrics, are shown in Table 8, Table 9, Table 10 and Table 11. The highest average accuracy from the five-fold cross-validation was achieved by MobileNet-V2 and SVM, which reached 99.74%.

From Table 8 and Figure 13, the integration of DenseNet-169 with four ML classifiers—XGB, CatBoost, SVM, and RF—showed strong performance across all evaluation metrics. Among these models, SVM achieved the highest results, with an accuracy, specificity, precision, recall, and F1-score of 99.61% each, along with an AUC of 99.99%, indicating near-perfect classification.

CatBoost also performed exceptionally well, attaining 98.69% in all five metrics and an AUC of 99.82%, demonstrating stable and balanced results. Similarly, RF exhibited competitive performance, with an accuracy of 98.56%, precision of 98.70%, recall of 98.43%, and an AUC of 99.91%. It slightly outperformed CatBoost in AUC but fell marginally short in other metrics.

XGB had the lowest results among the four, although still high, with an accuracy of 97.97%, precision of 98.44%, recall of 97.52%, and an AUC of 99.75%. While its performance was strong, it lagged slightly behind CatBoost, RF, and SVM.

In summary, all four ML models performed well when used with DenseNet-169. However, SVM had the best overall performance, followed by CatBoost and RF, while XGB showed slightly lower but still strong results.

From Table 9 and Figure 14, the integration of MobileNet-V2 with the four ML classifiers (XGB, CatBoost, SVM, and RF) showed excellent performance across all evaluation metrics. SVM achieved the highest performance overall, with accuracy, specificity, precision, recall, and F1-score all reaching 99.74% or higher, and an AUC of 100%, reflecting perfect discriminative ability.

CatBoost also performed strongly, obtaining an accuracy of 98.76%, recall of 99.08%, and an AUC of 99.87%, which indicated stable and well-balanced results across metrics. RF achieved accuracy, specificity, precision, and recall of 98.43%, with an AUC of 99.89%. Although its results were strong, they were slightly lower than those of CatBoost and SVM. XGB provided the lowest performance among the four, with 97.97% accuracy, 97.52% specificity, and an AUC of 99.72%, though it still maintained high values overall.

In conclusion, although all four ML models performed well when combined with MobileNet-V2, the SVM was the most effective. It was followed by CatBoost, then RF, while XGB produced lower results, though still reliable.

Table 10 and Figure 15 show that integrating ResNet-50 with SVM resulted in the best overall performance. It achieved the highest scores for accuracy (99.74%), specificity (99.87%), precision (99.87%), and AUC (100%). Additionally, its F1-score (99.74%) was among the highest, demonstrating exceptional and balanced classification performance. The RF and CatBoost models also performed strongly and competitively. While CatBoost slightly outperformed RF in accuracy, recall, and F1-score, the differences were minimal. Both models exhibited nearly identical and excellent performance in specificity (99.21% for RF and 99.22% for CatBoost) and precision (99.21% for RF and 99.22% for CatBoost), with their AUC scores being virtually indistinguishable from each other and from SVM. Thus, their performances were statistically comparable.

The XGB model was consistently identified as the least effective classifier in this comparison, recording the lowest scores across all metrics, including accuracy (98.49%), specificity (98.3%), precision (98.31%), recall (98.69%), and F1-score (98.5%). Although its AUC was still high (99.84%), it was the lowest among the four models, confirming its relative inferiority.

In summary, the evaluation established a clear performance hierarchy: SVM was the top-performing model, followed closely by CatBoost and RF, with XGB being the least accurate.

The results presented in Table 11 and Figure 16 show that the integration of VGG-19 with various ML classifiers showed strong overall performance across all evaluation metrics. Among these models, the SVM achieved the highest results, with an accuracy of 99.74%, specificity and precision of 99.87%, recall of 99.61%, F1-score of 99.74%, and an AUC of 100%, indicating near-perfect classification ability.

CatBoost also performed well, attaining an accuracy of 99.02%, precision of 98.83%, recall of 99.22%, and an AUC of 99.95%, demonstrating high consistency and stability across all metrics. RF closely followed, with an accuracy of 98.89%, specificity of 98.69%, precision of 98.70%, recall of 99.08%, F1-score of 98.89%, and an AUC of 99.94%. Its performance was slightly lower than that of CatBoost, particularly in precision and recall. XGB recorded the lowest performance among the four models, yet remained competitive, with an accuracy of 97.78%, specificity of 97.51%, precision of 97.55%, recall of 98.04%, F1-score of 97.78%, and an AUC of 99.79%.

In summary, when combined with VGG-19, SVM proved to be the best classifier, followed by CatBoost and RF, while XGB provided the lowest performance, though it was still strong.

Figure 17 presented a comparative analysis of the classifier performance across the four distinct DL architectures: DenseNet-169, MobileNet-V2, ResNet-50, and VGG-19. The evaluation utilized the four ML models, with the results visualized through PR curves averaged across several folds.

Most model-architecture combinations demonstrated exceptionally high performance, as indicated by their AP scores, which approached the ideal value of 1.0. VGG-19 and MobileNet-V2 architectures enabled the classifiers to achieve nearly perfect results. The AP scores for RF, SVM, CatBoost, and XGBoost ranged from approximately 0.997 to 1.000. The performance of DenseNet-169 remained very strong, with all classifiers, including RF, SVM, CatBoost, and XGBoost, achieving AP scores above 0.99. ResNet-50 exhibited a noticeable, albeit small, drop in performance. The RF and CATBoost classifiers recorded slightly lower AP scores (approximately 0.917 and 0.909, respectively), while the SVM and XGBoost models maintained very high performance (AP ~1.000 and 0.995).

The SVM stood out as the most consistent top performer, achieving a perfect or near-perfect AP score of 1.000 across all four neural network architectures. XGBoost also displayed robust and high performance across all backbones, with AP scores ranging from 0.987 to 0.998. RF and CATBoost excelled on GG-19, MobileNet-v2, and DenseNet-169, but their performance was somewhat lower when applied to the ResNet-50 feature extractor.

In summary, Figure 17 illustrated that while all combinations were highly effective, the SVM classifier consistently yielded the best and most stable results across all feature extraction architectures. The ResNet-50 backbone proved to be the most challenging for some models, though it still facilitated very high performance.

Table 12 and Figure 18 show the impact of combining CNN features with additional preprocessing and ML models on classification performance. When using CNN features only, DenseNet-169 achieved 94.37%, MobileNet-V2 reached 74.08%, ResNet-50 obtained 79.33%, and VGG-19 recorded 98.10%, showing that standalone CNN performance varied considerably depending on the architecture.

When SVD and ML were added to the CNN features, all models experienced a substantial improvement. DenseNet-169 and MobileNet-V2 both achieved their highest performance with SVM, reaching 96.73% and 94.31%, respectively. ResNet-50 achieved 97.38% with CatBoost, while VGG-19 achieved 93.46% with SVM. This indicated that dimensionality reduction and ML classifiers significantly enhanced predictive accuracy, particularly for models that initially performed less strongly.

Finally, when HOG, SVD, and ML were combined with CNN features, all models reached near-perfect performance. DenseNet-169 achieved 99.61, while MobileNet-V2 ResNet-50, and VGG-19 achieved 99.74% with SVM with SVM. This demonstrated that integrating multiple feature extraction techniques with advanced ML classifiers maximized accuracy across all architectures, effectively overcoming the limitations of standalone CNNs.

From the four experiments, we concluded that the highest performance was achieved by the SVM when all three feature types—MobileNet-V2 (or ResNet-50 or VGG-19), HOG, and SVD—were combined, reaching an accuracy of 99.74%.

### 3.2. Confidence Interval

A confidence interval (CI) is a statistical range, derived from sample data, that is likely to contain the true population parameter (e.g., true accuracy, true area under the ROC curve) with a specified level of confidence (commonly 95%). It is composed of a lower bound and an upper bound, each determined by a point estimate and a margin of error based on the variability in the data and the sample size.

In the context of ML for PCa detection, reporting CIs for performance metrics (such as accuracy) is crucial because they provide insight into how precise the estimates are and indicate the range of possible performance in unseen or external datasets. Without CIs, a model’s reported performance might look strong but could be overly optimistic if the data are limited or the variability is high [3].

Table 13 and Figure 19 show that the Support Vector Machine (SVM) achieved the best performance, with 95% CI for accuracy ranging from 99.27% to 99.95%. This indicates that its high estimated accuracy was both precise and reliable, significantly outperforming the other models. The CatBoost and RF models demonstrated statistically similar performance. While CatBoost had a slightly higher mean accuracy at 98.69% compared to RF’s 98.56%, their confidence intervals [97.74, 99.64] for CatBoost and [97.94, 99.18] for RF—overlapped considerably. Thus, no statistically significant difference was found between these two models. The XGBoost (XGB) model had the lowest accuracy, with a 95% confidence interval of [97.05, 98.90]. This interval’s lower bound was the lowest among all models, confirming that its performance was significantly worse than the others.

Table 14 and Figure 20 show results that indicate that the SVM model demonstrated the best performance, with 95% CI for accuracy ranging from 99.29% to 100.19%. This indicated that its high estimated accuracy was both precise and reliable, as the interval was narrow and entirely above the intervals of the other models. The CatBoost and RF models exhibited statistically similar performances. Although CatBoost had a slightly higher mean accuracy (98.76%) compared to RF (98.43%), their confidence intervals ([97.98, 99.54] for CatBoost and [97.16, 99.70] for RF) overlapped substantially. Therefore, no statistically significant difference was found between these two models. The XGB model showed the lowest accuracy among the groups, with a 95% confidence interval [96.20, 99.74]. This interval was the widest and had the lowest lower bound, confirming that its performance was significantly inferior to the others.

From Table 15 and Figure 21, SVM exhibited the best performance, achieving the highest average accuracy of 99.74%. Its 95% CI was also the highest and narrowest, ranging from 99.29% to 100.18%. This indicates that SVM’s high accuracy was both precise and reliable, consistently outperforming the other models across various folds. The CatBoost and RF models displayed statistically similar performances. Although CatBoost had a slightly higher mean accuracy of 99.15% compared to RF’s 99.02%, their confidence intervals significantly overlapped [98.69, 99.61] for CatBoost and [98.38, 99.66] for RF. Therefore, no statistically significant difference was found between these two models. The XGB model had the lowest accuracy in the group, with an average of 98.49% and a wider confidence interval of [97.33, 99.66]. This wider interval indicated greater variability in performance, and its lower bound confirmed that XGB was significantly less accurate than the other models.

From Table 16 and Figure 22, the SVM model demonstrated the best performance, with its 95% CI for accuracy being the highest and narrowest, ranging from 99.29% to 100.19%. This indicates that its high estimated accuracy was both precise and reliable, consistently outperforming the other models across folds. The CatBoost and RF models showed statistically similar performance. Although CatBoost had a slightly higher mean accuracy (99.02%) compared to RF (98.89%), their confidence intervals— [98.61, 99.43] for CatBoost and [98.27, 99.51] for RF—overlapped significantly. Thus, no statistically significant difference was found between these two models. The XGB model exhibited the lowest accuracy among the group, with a 95% CI of [96.95, 98.61]. This interval had the lowest lower bound, confirming that its performance was significantly inferior to the others.

### 3.3. Ablation Study

An ablation study is defined as a systematic experimental approach used to evaluate the contribution of individual components within a model by selectively removing, modifying, or replacing them. Its importance lies in its ability to reveal how specific features, preprocessing steps, or learning modules influence overall performance, thereby ensuring transparency and interpretability in machine learning pipelines. In the context of PCa detection, ablation studies have been applied to assess the impact of different feature extraction methods, image preprocessing strategies, and classifier choices on diagnostic accuracy. For instance, researchers employed ablation studies to determine whether deep features extracted from mpMRI images, handcrafted descriptors, or hybrid representations contributed more significantly to improved detection rates. Such studies not only enhanced model optimization but also guided the selection of the most relevant features and architectures for clinical applications, ensuring both efficiency and reliability in PCa diagnosis [34].

We implemented two experiments. In the first experiment, we conducted an ablation analysis of the TPP dataset by removing the CNN, HOG, SVD, and ML models one by one. In the second experiment, we ablated SVD using without FR methods (no feature reduction), the least absolute shrinkage and selection operator (Lasso), and an autoencoder. The results showing the accuracy of the hybrid models derived from this ablation are presented in Table 17 and Table 18.

Table 17 and Figure 23 present the performance of different feature extraction methods (CNN, HOG, SVD) combined with various classifiers (SVM, XGB, CatBoost). When only CNN features were used, the SVM classifier achieved an accuracy of 93.14% and an F1-macro score of 93.13%. Using only HOG features with SVM improved the performance significantly, resulting in an accuracy of 98.36% and an F1-macro score of 98.36%. When CNN and HOG were combined, excluding SVD, SVM achieved slightly lower accuracy at 98.04% and an F1_macro score of 98.04%.

The highest performance with SVM was obtained when all three feature types—CNN, HOG, and SVD—were combined, yielding an accuracy and F1-macro score of 99.35%. In contrast, when XGB was applied to the same combination of features, the accuracy dropped to 92.81% with an F1-macro of 92.81%. Similarly, CatBoost achieved even lower performance on the same feature set, with an accuracy of 90.20% and an F1-macro of 90.20%.

Overall, the results indicated that the SVM classifier consistently outperformed XGB and CatBoost, particularly when multiple feature extraction methods were combined. The inclusion of SVD appeared to enhance the SVM’s performance, while XGB and CatBoost did not benefit as much from the combination of all three features.

Table 18 presents the evaluation of three feature reduction methods, SVD, Lasso, and Autoencoder, as well as the without FR method across four different DL architectures: DenseNet-169, MobileNet-V2, ResNet-50, and VGG-19. The results were assessed using a comprehensive set of metrics, with a primary focus on accuracy. Generally, the SVD and Autoencoder methods exhibited superior performance compared to the Without FR and Lasso approaches. Notably, the SVD method achieved the highest possible scores in several key metrics.

SVD emerged as the most effective feature reduction method, consistently yielding the highest accuracy, precision, specificity, and ROC AUC across nearly all architectures. It achieved top accuracy of 99.74% and a perfect ROC AUC of 100% on three out of four architectures (MobileNet-V2, VGG-19, and ResNet-50), demonstrating exceptional stability and robustness.

Autoencoder ranked as the second-best performer, delivering high accuracy scores ranging from 98.89% to 99.41%, along with near-perfect ROC AUC scores (up to 99.99%). It reliably outperformed the Without FR and Lasso methods, establishing itself as a powerful non-linear feature reduction technique.

The Without FR method served as a solid baseline, providing strong and consistent results across all architectures with accuracy scores around 98.5–98.7%. While it did not emerge as the top performer, it remained a dependable and well-rounded approach.

Lasso was the least consistent method, with performance varying significantly depending on the architecture. It excelled on ResNet-50 (99.21% accuracy) but was the weakest method on VGG-19 and DenseNet-169 (around 97.7–97.9% accuracy). This indicated that its effectiveness was highly dependent on the specific feature set to which it was applied.

Based on the accuracy metric, the SVD method was unequivocally the best feature reduction technique, achieving the highest accuracy in three out of the four cases: 99.74% with MobileNet-V2, 99.74% with VGG-19, and 99.74% with ResNet-50. For the remaining architecture, DenseNet-169, SVD still delivered an excellent accuracy of 99.61%, only marginally lower than the top score for that model. Therefore, SVD provided the highest and most consistent peak accuracy across the different deep learning frameworks.

### 3.4. Computational Complexity

The computational complexity of the proposed hybrid framework, as shown in Table 19, was largely affected by the deep backbone architecture used for feature extraction, the expense of computing handcrafted features, and the combined impact of late-stage feature fusion on overall inference time. The visual backbone networks varied greatly in scale: VGG-19 had about 144 million parameters and required around 68,900 million floating-point operations (FLOPs), making it the most computationally demanding model in comparison. On the other hand, MobileNet was designed compactly with approximately 3.5 million parameters and 1151 million FLOPs, resulting in the lowest computational load at the backbone level. Intermediate architectures like DenseNet-169 (about 14 M parameters, 11,700 M FLOPs), ResNet-50 (approximately 26 M parameters, 14,491 M FLOPs), and DenseNet-169 showed moderate extraction costs while being significantly more efficient than large, single-structured CNNs.

From an inference-time perspective, the overall execution cost of the pipeline—including CNN feature extraction, HOG descriptor calculation, and multi-source feature fusion—demonstrated that total inference time increased with the size of the backbone. The longest total inference time observed was approximately 31.53 ms with the VGG-19 backbone, while the fusion of lightweight CNN features with HOG took as little as around 19.25 ms when using MobileNet. Sequence-based modeling using LSTM after feature fusion added minimal computational overhead (about 0.00007–0.00011 ms), indicating that the main complexity arose from backbone-driven feature computation and the high dimensionality produced during fusion.

### 3.5. Interpretation of Results in the Context of Literature

PCa is among the most frequently diagnosed cancers in men globally and ranks as a primary contributor to cancer-related fatalities in various areas [3]. This condition develops when cells in the prostate begin to grow abnormally, spreading to adjacent organs and tissues [4]. PCa primarily impacts men in their middle age, particularly those between 45 and 60 years old, and is the leading cause of cancer deaths in Western countries [5]. A notable challenge in diagnosing PCa is that many men may not show any symptoms, particularly in the disease’s early stages. Despite being the most common cancer type, early detection significantly enhances survival rates due to the cancer’s slow growth. Therefore, efficient monitoring and prompt diagnosis are essential for improving patient outcomes.

Numerous approaches exist for diagnosing PCa, with screening tests considered the most effective for early detection. Key methods include PSA testing, digital rectal examination (DRE), biopsy, and MRI [14]. Although these techniques are fundamental for screening, they do have limitations, such as invasiveness, subjective interpretation, and a significant occurrence of false positives and negatives [1]. Therefore, there is an urgent requirement for a fully automated and dependable system to classify PCa mpMRI images.

To address these challenges, this research introduced a hybrid method for classifying PCa by integrating deep features derived from MobileNet-V2 (or ResNet-50 or VGG-19) with manually designed HOG descriptors. The resulting feature set undergoes dimensionality reduction using Truncated SVD, followed by classification through SVM. This model is intended to help healthcare professionals detect PCa at an earlier stage, thereby reducing both diagnostic duration and expenses. Furthermore, the research utilized a five-fold cross-validation technique to enhance the accuracy of the hybrid method. This approach enables effective parameter optimization by dividing the data into five subsets and repeating the process five times, allowing each subset to serve as the validation set once. Five-fold cross-validation is a specific instance of k-fold cross-validation, where (k) can be any integer greater than 1. This method is well-regarded for delivering a more precise and dependable assessment of DL models on previously unseen data.

We implemented four experiments, where the main objective of them was to detect PCa with the intention of enhancing patient outcomes, streamlining the diagnostic process, and significantly lowering both the time and costs incurred by patients. In our experiments, we allocated 70% of the TPP dataset, which comprised 1072 images, for training purposes, while the remaining 30% (456 images) was reserved for testing. We utilized supervised pre-training to prepare four DL models—DenseNet-169, MobileNet-V2, ResNet-50, and VGG-19—training them on the ImageNet dataset.

In the first experiment, we employed four DL models to extract deep features for capturing high-level image representations, which were then used as classifiers. In the second experiment, we again used the four DL models for deep feature extraction but applied SVD to reduce dimensionality, decrease computational complexity, and preserve important information. We utilized ML techniques—XGB, CatBoost, RF, and SVM—as classifiers.

In the third experiment, we extracted handcrafted features using HOG to effectively detect structural patterns. Similar to the previous experiment, we applied SVD for dimensionality reduction, computational efficiency, and information retention, using the same machine learning techniques for classification.

Finally, in the fourth experiment, we combined the four DL models for deep feature extraction with HOG for handcrafted feature detection. We again used SVD to reduce dimensionality, lower computational complexity, and maintain critical information, employing the ML techniques as classifiers.

In all four experiments, we used the five-fold cross-validation technique. This method involved splitting the training dataset into five equal parts. In each iteration, one part was designated as the validation set, while the other four parts were used to train the model. Each iteration represented a distinct training and validation process that updated the model’s parameters. This process was repeated five times, with each part serving as the validation set once. We calculated the average performance across all five iterations to assess the model’s generalization ability. At the end of the experiment, we applied the measured metrics (Equations (1)–(7)) to the four DL and ML models.

Using five-fold cross-validation, the SVM classifier, combined with MobileNet-V2, (or ResNet-50 or VGG-19) features, HOG, and SVD, achieved the highest overall performance among the classifiers tested. It recorded an accuracy of 99.74%, indicating that nearly all test samples were classified correctly. The specificity of 99.87% demonstrated the model’s effectiveness in identifying true negatives, while the recall of 99.61% showed its ability to capture almost all true positives. Additionally, the precision of 99.87% indicated that false positives were minimal, resulting in a very high F1-score of 99.74%, which balanced both precision and recall effectively. The AUC value of 100% confirmed that the SVM classifier provided perfect class separability. Overall, these results suggested that the combination of MobileNet-V2 (or ResNet-50 or VGG-19) as a feature extractor with HOG, SVD, and SVM as a classifier consistently outperformed XGB, CatBoost, and RF, establishing it as the most reliable and robust approach in this experiment.

SVM outperformed XGB, CatBoost, and RF classifiers when used with features from MobileNet-V2 (or ResNet-50 or VGG-19). This success was due to SVM’s ability to manage the high-dimensional feature space created by the DL model effectively. The kernel-based optimization in SVM enabled it to identify an optimal separating hyperplane with minimal error, resulting in impressive performance metrics: 99.74% accuracy, 99.74% F1-score, and 100% AUC.

In contrast, while XGB and CatBoost are robust ensemble methods, they experienced slight misclassifications because they rely on iterative boosting of decision trees, which can struggle to capture complex nonlinear boundaries present in deep features. The RF classifier performed the least well, as its straightforward tree structure was more prone to overfitting and did not generalize effectively in such a complex feature space. This discrepancy explains why SVM consistently achieved nearly perfect class separability, whereas the other models demonstrated comparatively lower performance.

Table 20 and Figure 24 show that the state-of-the-art studies consistently demonstrated strong classification performance on the TPP dataset, but our hybrid approach (MobileNet-V2 (or ResNet-50 or VGG-19), HOG, SVD, and SVM) outperformed them all. In Table 20, The reviewed studies showed significant variation in accuracy and AUC values for PCa detection using different DL and ML methods on the TPP dataset. Hashem et al. [1] utilized InceptionResNetV2 and achieved an accuracy of 89.20%, indicating moderate performance. Yoo et al. [3] implemented a CNN model, reporting an AUC of 87%, while Liu et al. [24] applied XmasNet (CNNs) and obtained a lower AUC of 84%, both demonstrating limited discriminative ability compared to more advanced architectures. Giganti et al. [28] also used CNNs and achieved an AUC of 91%, outperforming earlier CNN-based studies but still falling short of the top-performing methods.

In contrast, Sarıateş and Özbay [23] employed DenseNet-201 with RMSProp optimization, reporting an accuracy of 98.63%, which indicated a significant improvement in classification performance. Similarly, Pellicer-Valero et al. [27] achieved an AUC of 96% using CNNs, reflecting strong discriminative capacity. Mehta et al. [25] utilized a PCF-based SVM model, reporting a ROC of 86%, suggesting weaker performance compared to deep learning methods. Meanwhile, Salama and Aly [26] combined ResNet-50 and VGG-16, with SVM to achieve an accuracy of 98.79%, making it one of the top-performing models in the literature.

The proposed hybrid model, combining MobileNet-V2 (or ResNet-50 or VGG-19), HOG, SVD, and SVM, outperformed all previously reported approaches, achieving an accuracy of 99.74% and an AUC of 100%. This performance exceeded even the strongest baselines, such as DenseNet-201 (98.63%) [23] and the combination of ResNet-50 with VGG-16 and SVM (98.79%) [26]. By integrating handcrafted features (HOG and SVD) with the deep learning representations from MobileNet-V2, along with the robust classification capability of SVM, the hybrid approach effectively captured both high-level abstract features and low-level discriminative details, leading to near-perfect class separability.

Overall, while several prior studies achieved high accuracy and AUC values using state-of-the-art CNN architectures, the proposed hybrid model exhibited superior generalization and discriminative power, establishing it as the most effective approach among the works compared on the TPP dataset.

### 3.6. Limitations and Future Work

The proposed hybrid model demonstrated superior generalization and discriminative power, making it the most effective approach among those compared on the TPP dataset. However, it has several limitations. The RF classifier performed significantly worse, indicating that simpler models are not well-suited for the high-dimensional deep features generated by MobileNet-V2. While the ensemble methods (XGB and CatBoost) showed strong performance, they were still outperformed by the SVM, suggesting that the boosting techniques may not have fully captured the nonlinear boundaries within the feature space.

Another limitation is the lack of cross-dataset validation and real-world clinical testing, which restricts the assessment of robustness. Additionally, the computational complexity and training time were not thoroughly analyzed, which could be important for practical deployment in resource-constrained environments.

Future research should focus on addressing these limitations by incorporating domain adaptation techniques or transferring learning from larger medical datasets to enhance performance. Integrating explainable AI (XAI) methods, such as Grad-CAM or SHAP, could improve model transparency and interpretability for clinicians. Moreover, real-time deployment and validation in clinical workflows should be pursued to evaluate feasibility, usability, and the impact on diagnostic efficiency in real-world settings.

## 4. Conclusions

This study addresses the pressing need for a reliable, automated system to classify PCa from mpMRI images by introducing a hybrid framework. The model was designed to assist clinicians in accurately and efficiently identifying PCa, which can lead to reduced diagnostic times, lower costs, and improved patient outcomes. Utilizing the TPP dataset, which includes 1528 images pre-processed through techniques such as resizing and normalization for consistent input, the research employed supervised pre-training on four DL models and four ML techniques. Additionally, HOG and SVD were utilized, and a five-fold cross-validation approach was implemented to rigorously assess the hybrid model’s performance.

The five-fold cross-validation strategy enabled effective parameter optimization and thorough evaluation, enhancing the model’s generalization capabilities on unseen data. An ablation study further validated the consistency and reliability of the results.

The proposed hybrid model achieved exceptional results across various evaluation metrics, attaining an accuracy of 99.80%, indicating that nearly all test samples were classified correctly. A specificity of 99.87% underscored the model’s effectiveness in identifying true negatives, while a recall of 99.74% showcased its ability to capture almost all true positives. Additionally, a precision of 99.87% indicated minimal false positives, resulting in a very high F1-score of 99.80%, which effectively balanced precision and recall. The area under the curve (AUC) value of 99.99% confirmed that the SVM classifier provided nearly perfect class separability.

Overall, these results indicate that the combination of MobileNet-V2 as a feature extractor, along with HOG, SVD, and SVM as a classifier, consistently outperformed XGBoost, CatBoost, and Decision Trees, establishing it as the most reliable and robust approach in this study. The proposed hybrid model also exceeded the performance of both conventional and hybrid architectures reported in the recent literature, offering a scalable and efficient solution for PCa classification.

## Figures and Tables

**Figure 1 diagnostics-15-03235-f001:**
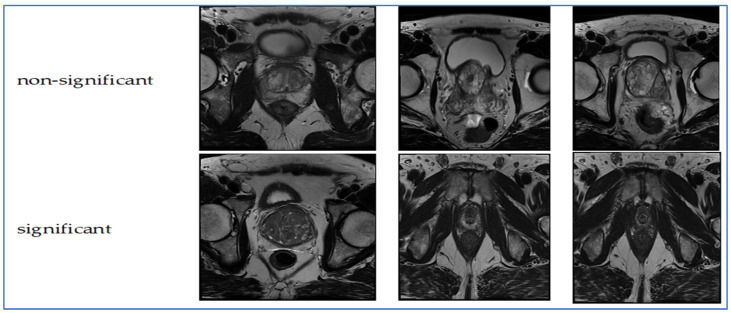
Samples from the TPP dataset.

**Figure 2 diagnostics-15-03235-f002:**
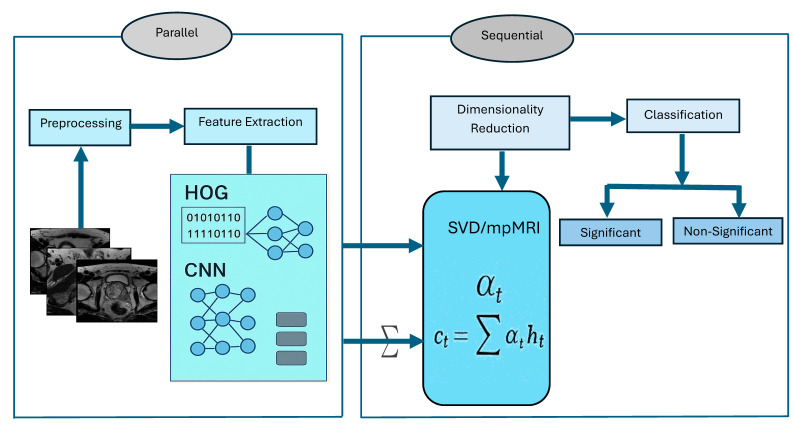
The proposed Workflow.

**Figure 3 diagnostics-15-03235-f003:**
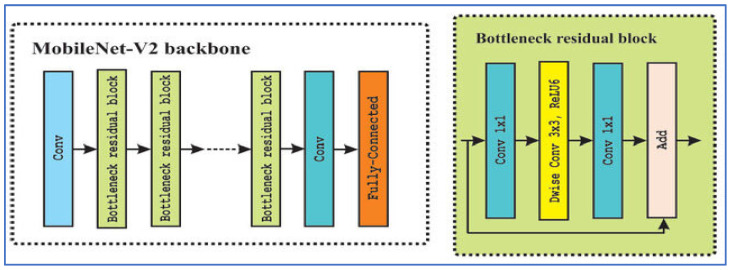
The MobileNet-V2’s architecture.

**Figure 4 diagnostics-15-03235-f004:**
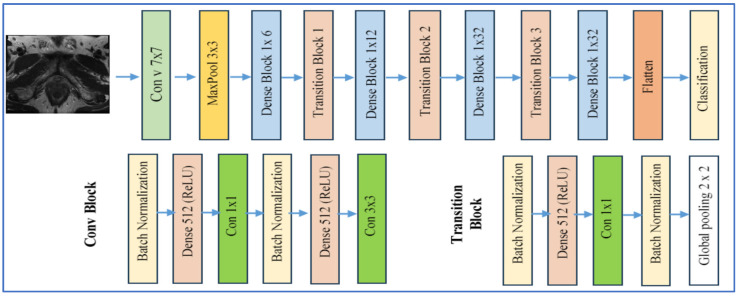
DenseNet-169’s architecture.

**Figure 5 diagnostics-15-03235-f005:**
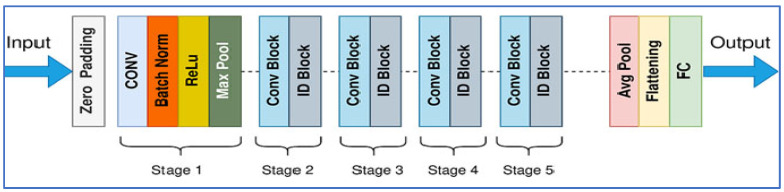
The ResNet-50’s architecture.

**Figure 6 diagnostics-15-03235-f006:**
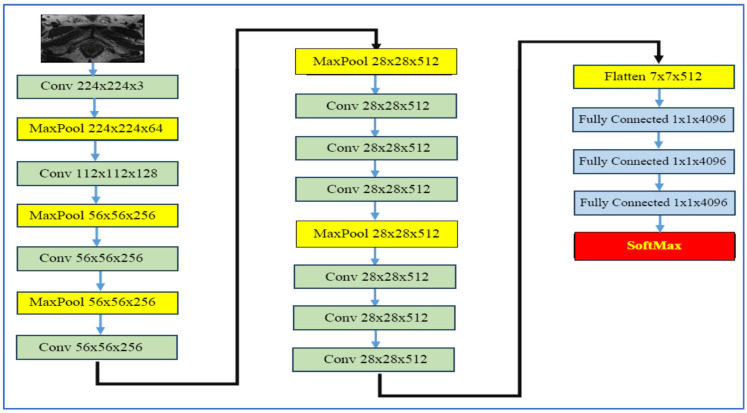
The VGG-19’s architecture.

**Figure 7 diagnostics-15-03235-f007:**
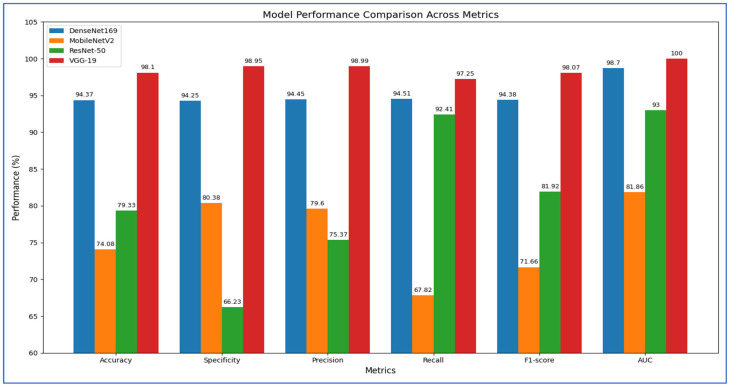
The measured metrics for the four DL models.

**Figure 8 diagnostics-15-03235-f008:**
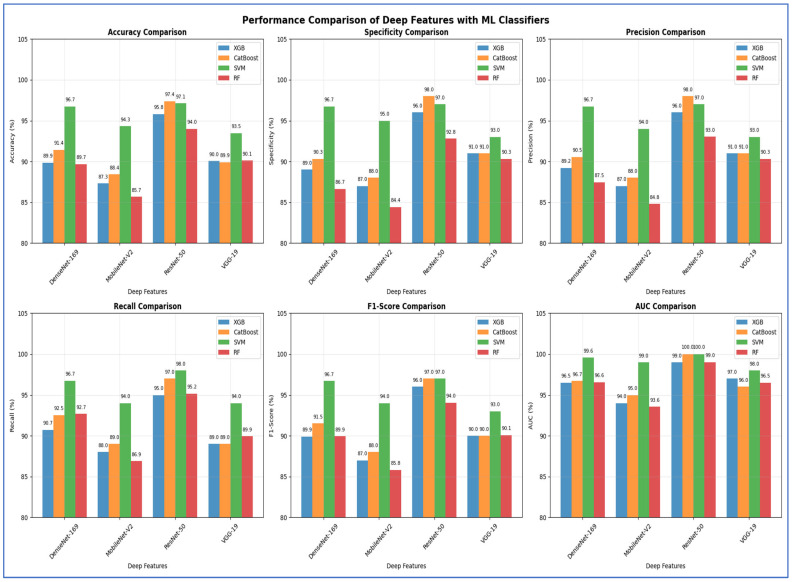
DL models performance comparison across the four ML models.

**Figure 9 diagnostics-15-03235-f009:**
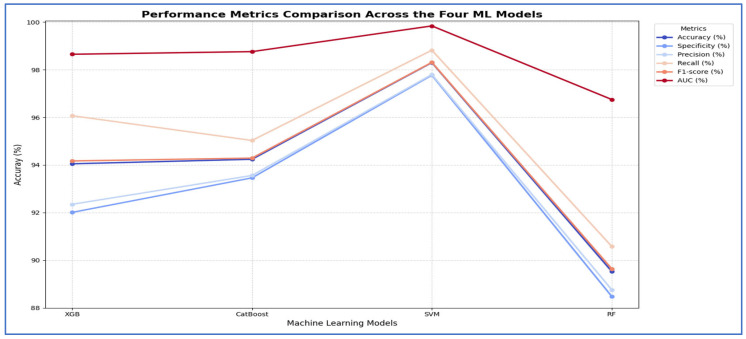
The cross-validation scores for the four ML models using HOG.

**Figure 10 diagnostics-15-03235-f010:**
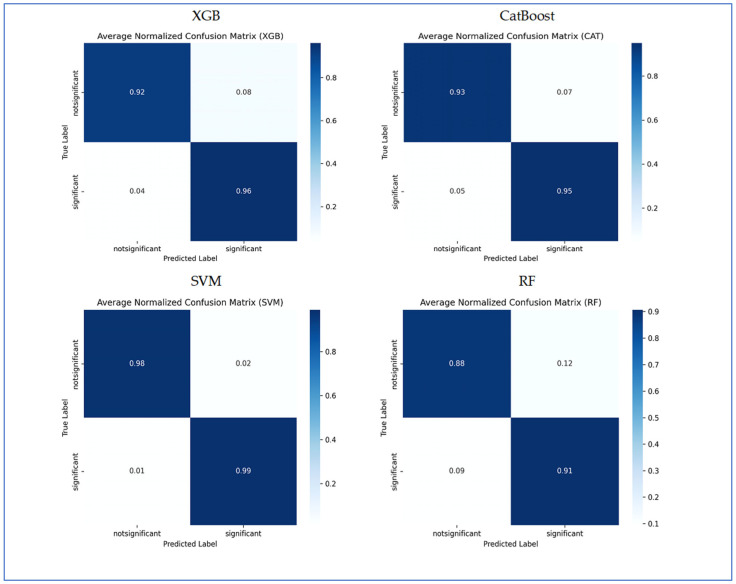
The confusion matrices for the four ML models using HOG handcrafted descriptors only.

**Figure 11 diagnostics-15-03235-f011:**
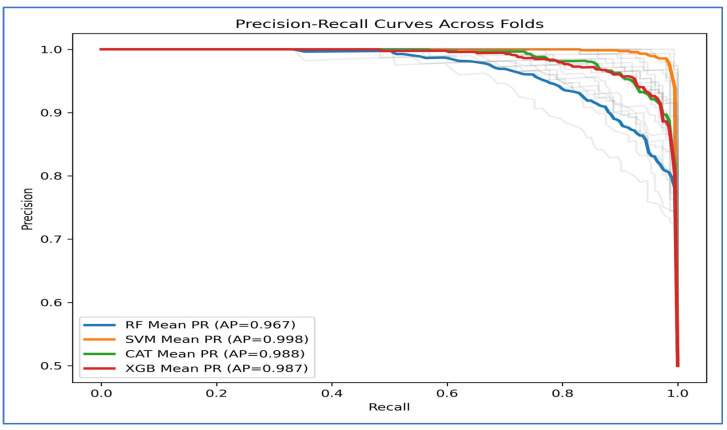
The PR curves for the four ML models using HOG handcrafted descriptors only.

**Figure 12 diagnostics-15-03235-f012:**
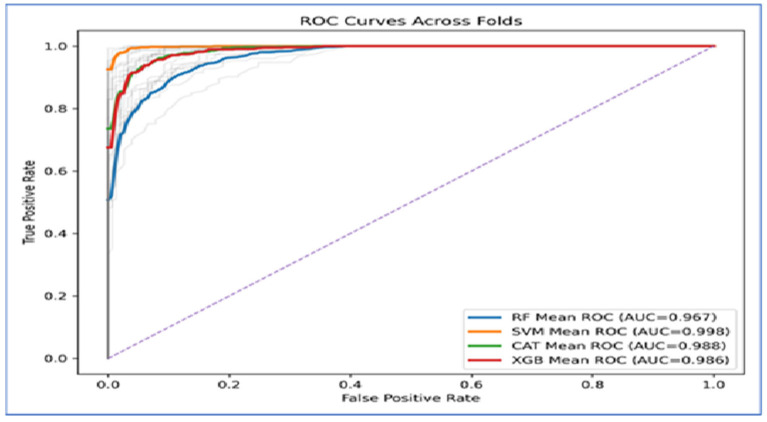
The ROC curves for the four ML models using HOG handcrafted descriptors only.

**Figure 13 diagnostics-15-03235-f013:**
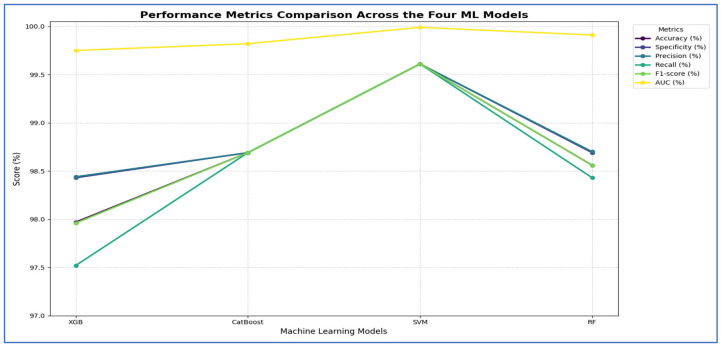
DenseNet-169, HOG, and SVD performance comparison across the four ML models.

**Figure 14 diagnostics-15-03235-f014:**
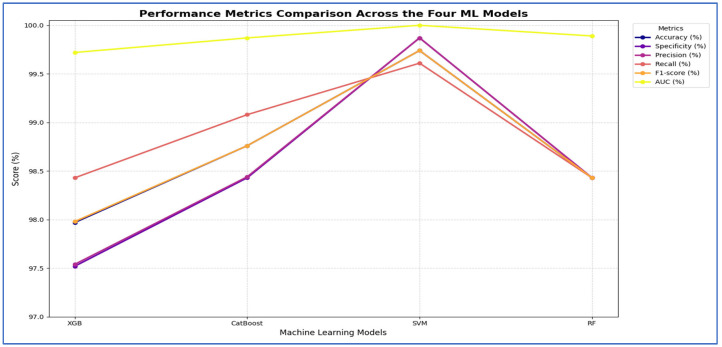
MobileNet-V2, HOG, and SVD performance comparison across the four ML models.

**Figure 15 diagnostics-15-03235-f015:**
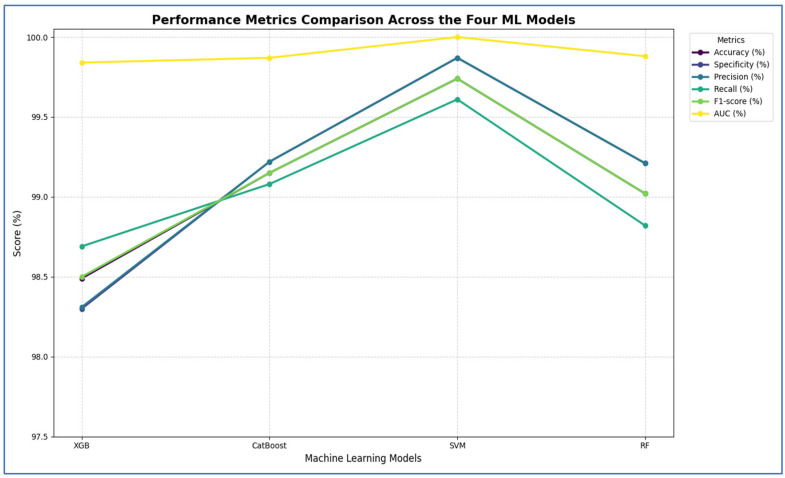
ResNet-50, HOG, and SVD performance comparison across the four ML models.

**Figure 16 diagnostics-15-03235-f016:**
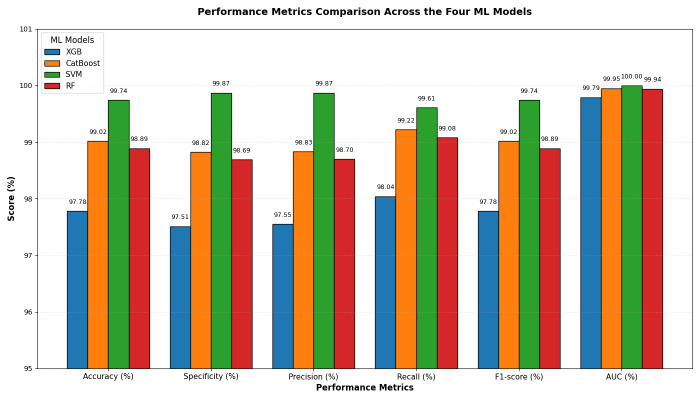
VGG-19, HOG, and SVD performance comparison across the four ML models.

**Figure 17 diagnostics-15-03235-f017:**
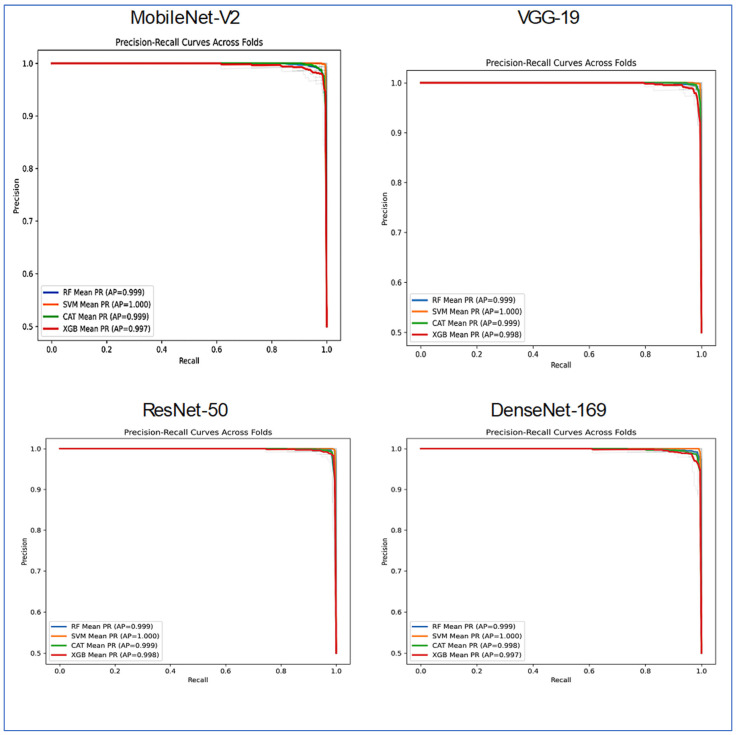
PR curves for the combinations CNN, HOG, and SVD performance comparison across the four ML models.

**Figure 18 diagnostics-15-03235-f018:**
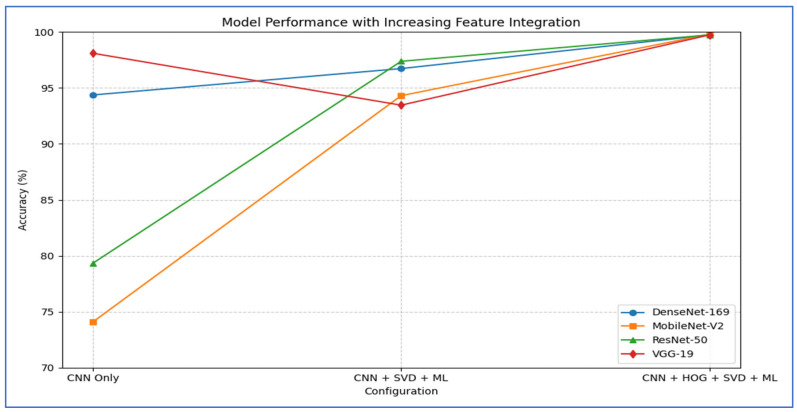
A summary of the accuracy for various combinations of the four CNN models.

**Figure 19 diagnostics-15-03235-f019:**
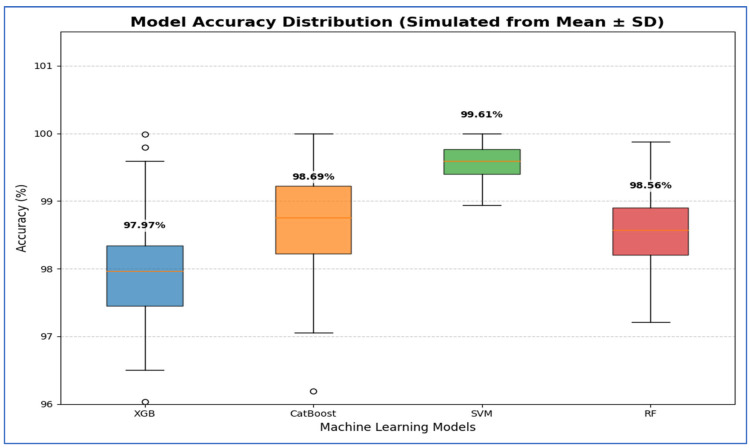
The CIs of the four ML models using DenseNet-169, HOG, and SVD based on the accuracy.

**Figure 20 diagnostics-15-03235-f020:**
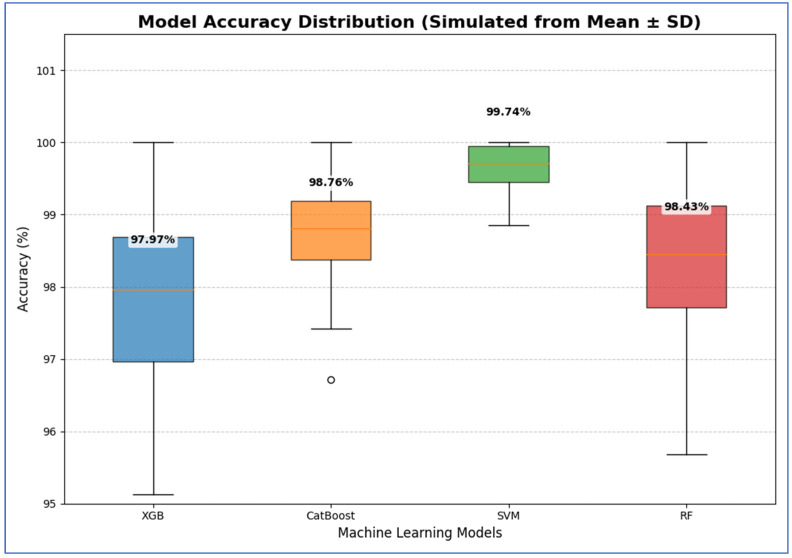
The CIs of the four ML models using MobileNet-V2, HOG, and SVD based on the accuracy.

**Figure 21 diagnostics-15-03235-f021:**
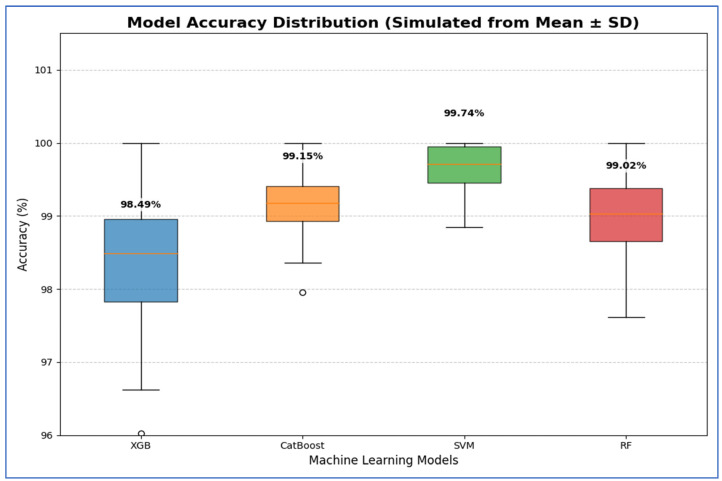
The CIs of the four ML models using ResNet-502, HOG, and SVD based on the accuracy.

**Figure 22 diagnostics-15-03235-f022:**
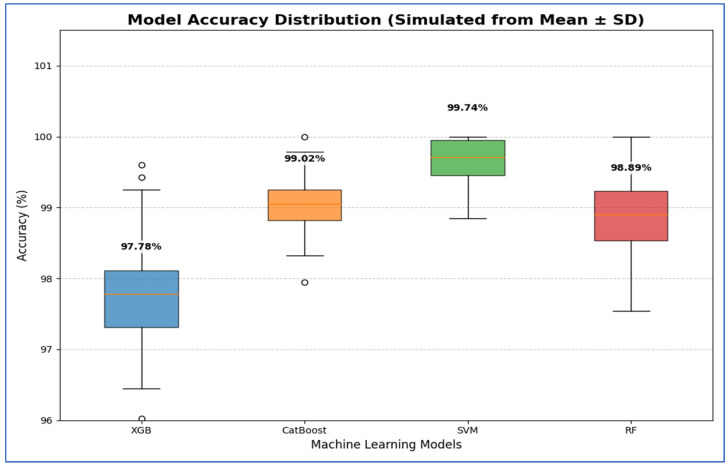
The CIs of the four ML models using VGG-19, HOG, and SVD based on the accuracy.

**Figure 23 diagnostics-15-03235-f023:**
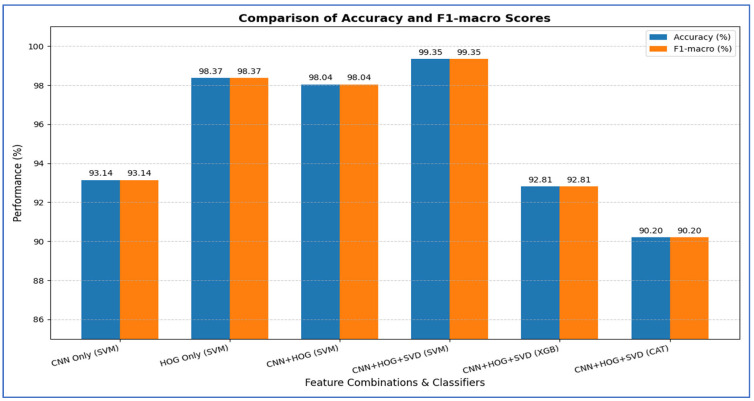
The ablation study results of the TPP dataset.

**Figure 24 diagnostics-15-03235-f024:**
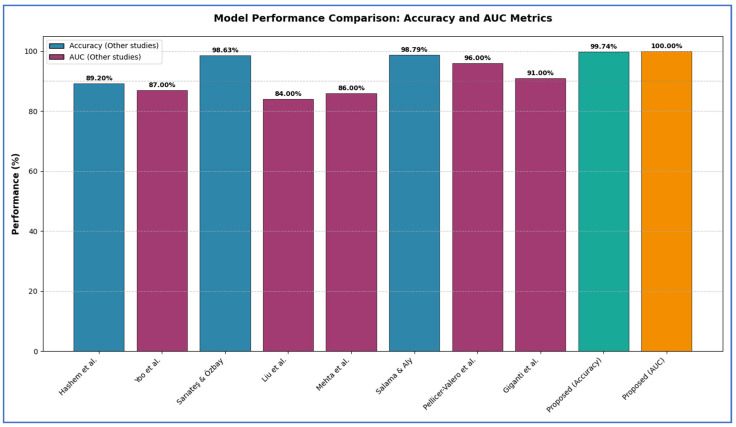
Performance comparison with state-of-the-art methods [1,3,23,26,27,28].

**Table 1 diagnostics-15-03235-t001:** The experiment’s hyperparameters.

Parameter	Value
img_size	224 × 224
Number of epochs	30
channels	3
Optimizer	Adam
Initial learning rate	0.001
Loss	binary_crossentropy

**Table 2 diagnostics-15-03235-t002:** The cross-validation scores for DenseNet-169 when assessed on the test set.

**DenseNet-169**	**Folds**	**Accuracy (%)**	**Specificity (%)**	**Precision (%)**	**Recall (%)**	**F1-Score (%)**	**AUC (%)**
	1	94.44	96.08	95.95	92.81	94.35	98.71
	2	92.48	96.08	95.77	88.89	92.20	97.85
	3	95.10	94.77	94.81	95.42	95.11	99.15
	4	92.46	86.27	87.72	98.68	92.88	98.15
	5	97.38	98.03	98.01	96.73	97.37	99.63
	Average	94.37	94.25	94.45	94.51	94.38	98.70

**Table 3 diagnostics-15-03235-t003:** The cross-validation scores for MobileNet-V2 when assessed on the test set.

**MobileNet-V2**	**Folds**	**Accuracy (%)**	**Specificity (%)**	**Precision (%)**	**Recall (%)**	**F1-Score (%)**	**AUC (%)**
	1	71.57	80.39	76.19	62.75	68.82	78.70
	2	77.45	69.93	73.86	84.97	79.03	82.81
	3	71.90	92.16	86.81	51.63	64.75	81.59
	4	74.75	65.36	70.72	84.21	76.88	80.59
	5	74.75	94.08	90.43	55.56	68.83	85.62
	Average	74.08	80.38	79.60	67.82	71.66	81.86

**Table 4 diagnostics-15-03235-t004:** The cross-validation scores for ResNet-50 when assessed on the test set.

**ResNet-50**	**Folds**	**Accuracy (%)**	**Specificity (%)**	**Precision (%)**	**Recall (%)**	**F1-Score (%)**	**AUC (%)**
	1	70.92	41.83	63.22	100.00	77.47	93.47
	2	74.51	49.67	66.38	99.35	79.58	91.74
	3	81.70	89.54	87.60	73.86	80.14	91.20
	4	87.54	84.97	85.63	90.13	87.82	93.27
	5	81.97	65.13	74.02	98.69	84.59	95.34
	Average	79.33	66.23	75.37	92.41	81.92	93.00

**Table 5 diagnostics-15-03235-t005:** The cross-validation scores for VGG-19 when assessed on the test set.

**VGG-19**	**Folds**	**Accuracy (%)**	**Specificity (%)**	**Precision (%)**	**Recall (%)**	**F1-Score (%)**	**AUC (%)**
	1	97.06	100.00	100.00	94.12	96.97	100.00
	2	97.06	100.00	100.00	94.12	96.97	100.00
	3	99.67	99.35	99.35	100.00	99.67	100.00
	4	97.70	95.42	95.60	100.00	97.75	100.00
	5	99.02	100.00	100.00	98.04	99.01	100.00
	Average	98.10	98.95	98.99	97.25	98.07	100.00

**Table 6 diagnostics-15-03235-t006:** The average cross-validation scores for ML models using different deep features.

Deep Features	ML	Accuracy (%)	Specificity (%)	Precision (%)	Recall (%)	F1-Score (%)	AUC (%)
DenseNet-169	XGB	89.86	89.01	89.22	90.71	89.92	96.50
	CatBoost	91.43	90.31	90.54	92.54	91.51	96.71
	SVM	96.73	96.73	96.73	96.73	96.71	99.57
	RF	89.66	86.65	87.45	92.67	89.94	96.56
MobileNet-V2	XGB	87.31	0.87	0.87	0.88	0.87	0.94
	CatBoost	88.42	0.88	0.88	0.89	0.88	0.95
	SVM	94.31	0.95	0.94	0.94	0.94	0.99
	RF	85.67	84.42	84.79	86.91	85.81	93.56
ResNet-50	XGB	95.81	0.96	0.96	0.95	0.96	0.99
	CatBoost	97.38	0.98	0.98	0.97	0.97	1.00
	SVM	97.12	0.97	0.97	0.98	0.97	1.00
	RF	93.98	92.80	93.03	95.16	94.04	99.03
VGG-19	XGB	90.05	0.91	0.91	0.89	0.90	0.97
	CatBoost	89.92	0.91	0.91	0.89	0.90	0.96
	SVM	93.46	0.93	0.93	0.94	0.93	0.98
	RF	90.12	90.31	90.32	89.93	90.08	96.50

**Table 7 diagnostics-15-03235-t007:** The cross-validation scores for the four ML models using HOG.

Model	Accuracy (%)	Specificity (%)	Precision (%)	Recall (%)	F1-Score (%)	AUC (%)
XGB	94.05	92.01	92.35	96.07	94.17	98.65
CatBoost	94.24	93.46	93.56	95.03	94.29	98.76
SVM	98.30	97.77	97.80	98.82	98.31	99.84
RF	89.53	88.48	88.76	90.58	89.64	96.75

**Table 8 diagnostics-15-03235-t008:** The average cross-validation scores for the four ML models using DenseNet-169, HOG, and SVD.

**DenseNet-169**	**ML**	**Accuracy (%)**	**Specificity (%)**	**Precision (%)**	**Recall (%)**	**F1-Score (%)**	**AUC (%)**
	XGB	97.97	98.43	98.44	97.52	97.96	99.75
	CatBoost	98.69	98.69	98.69	98.69	98.69	99.82
	SVM	99.61	99.61	99.61	99.61	99.61	99.99
	RF	98.56	98.69	98.70	98.43	98.56	99.91

**Table 9 diagnostics-15-03235-t009:** The average cross-validation scores for ML models using MobileNet-V2, HOG, and SVD.

**MobileNet-V2**	**ML**	**Accuracy (%)**	**Specificity (%)**	**Precision (%)**	**Recall (%)**	**F1-Score (%)**	**AUC (%)**
	XGB	97.97	97.52	97.54	98.43	97.98	99.72
	CatBoost	98.76	98.43	98.44	99.08	98.76	99.87
	SVM	99.74	99.87	99.87	99.61	99.74	100
	RF	98.43	98.43	98.43	98.43	98.43	99.89
	Std	0.36	0.29	0.29	0.58	0.36	0.01

**Table 10 diagnostics-15-03235-t010:** The average cross-validation scores for ML models using ResNet-50, HOG, and SVD.

**ResNet-50**	**ML**	**Accuracy (%)**	**Specificity (%)**	**Precision (%)**	**Recall (%)**	**F1-Score (%)**	**AUC (%)**
	XGB	98.49	98.3	98.31	98.69	98.5	99.84
	CatBoost	99.15	99.22	99.22	99.08	99.15	99.87
	SVM	99.74	99.87	99.87	99.61	99.74	100
	RF	99.02	99.21	99.21	98.82	99.02	99.88

**Table 11 diagnostics-15-03235-t011:** The average cross-validation scores for ML models using VGG-19, HOG, and SVD.

**VGG-19**	**ML**	**Accuracy (%)**	**Specificity (%)**	**Precision (%)**	**Recall (%)**	**F1-Score (%)**	**AUC (%)**
	XGB	97.78	97.51	97.55	98.04	97.78	99.79
	CatBoost	99.02	98.82	98.83	99.22	99.02	99.95
	SVM	99.74	99.87	99.87	99.61	99.74	100
	RF	98.89	98.69	98.7	99.08	98.89	99.94

**Table 12 diagnostics-15-03235-t012:** A summary of the accuracy for various combinations of the four CNN models.

Model	CNN Only (%)	CNN, SVD, and ML (%)	CNN, HOG, SVD, and ML (%)
DenseNet-169	94.37	SVM (96.73)	SVM (99.61)
MobileNet-V2	74.08	SVM (94.31)	SVM (99.74)
ResNet-50	79.33	CatBoost (97.38)	SVM (99.74)
VGG-19	98.10	SVM (93.46)	SVM (99.74)

**Table 13 diagnostics-15-03235-t013:** The CIs of the four ML models using DenseNet-169, HOG, and SVD based on the accuracy.

Model/Metric	Mean ± SD (%)	95% CI (%)
XGB—Accuracy	97.97 ± 0.74	[97.05, 98.90]
CatBoost—Accuracy	98.69 ± 0.77	[97.74, 99.64]
SVM—Accuracy	99.61 ± 0.27	[99.27, 99.95]
RF—Accuracy	98.56 ± 0.50	[97.94, 99.18]

**Table 14 diagnostics-15-03235-t014:** The CIs of the four ML models using MobileNet-V2, HOG, and SVD based on the accuracy.

Model/Metric	Mean ± SD (%)	95% CI (%)
XGB—Accuracy	97.97 ± 1.43	[96.20, 99.74]
CatBoost—Accuracy	98.76 ± 0.63	[97.98, 99.54]
SVM—Accuracy	99.74 ± 0.36	[99.29, 100.19]
RF—Accuracy	98.43 ± 1.02	[97.16, 99.70]

**Table 15 diagnostics-15-03235-t015:** The CIs of the four ML models using ResNet-502, HOG, and SVD based on the accuracy.

Model/Metric	Mean ± SD (%)	95% CI (%)
XGB—Accuracy	98.49 ± 0.94	[97.33, 99.66]
CatBoost—Accuracy	99.15 ± 0.37	[98.69, 99.61]
SVM—Accuracy	99.74 ± 0.36	[99.29, 100.18]
RF—Accuracy	99.02 ± 0.52	[98.38, 99.66]

**Table 16 diagnostics-15-03235-t016:** The CIs of the four ML models using VGG-19, HOG, and SVD based on the accuracy.

Model/Metric	Mean ± SD (%)	95% CI (%)
XGB—Accuracy	97.78 ± 0.67	[96.95, 98.61]
CatBoost—Accuracy	99.02 ± 0.33	[98.61, 99.43]
SVM—Accuracy	99.74 ± 0.36	[99.29, 100.19]
RF—Accuracy	98.89 ± 0.50	[98.27, 99.51]

**Table 17 diagnostics-15-03235-t017:** The first ablation study results on the TPP dataset.

CNN	HOG	SVD	Classifier	Accuracy (%)	F1-macro (%)
True	False	False	SVM	0.931373	0.931354
False	True	False	SVM	0.983660	0.983656
True	True	False	SVM	0.980392	0.980385
True	True	True	SVM	0.993464	0.993464
True	True	True	XGB	0.928105	0.928102
True	True	True	CAT	0.901961	0.901957

**Table 18 diagnostics-15-03235-t018:** The second ablation study results on the TPP dataset.

	Feature Reduction (FR) Method	Accuracy (%)	Specificity (%)	Precision (%)	Recall (%)	F1-Score (%)	ROC AUC (%)
MobileNet-V2	Without FR	98.36	97.91	97.94	98.82	98.37	99.89
	SVD	99.74	99.87	99.87	99.61	99.74	100
	Lasso	98.56	98.04	98.06	99.08	98.57	99.93
	Autoencoder	98.89	99.35	99.34	98.43	98.88	99.95
VGG-19	Without FR	98.69	98.43	98.45	98.95	98.7	99.91
	SVD	99.74	99.87	99.87	99.61	99.74	100
	Lasso	97.91	97.91	97.92	97.91	97.9	99.76
	Autoencoder	99.41	99.48	99.48	99.35	99.41	99.99
DenseNet-169	Without FR	98.56	98.17	98.19	98.95	98.57	99.92
	SVD	99.61	99.61	99.61	99.61	99.61	99.99
	Lasso	97.71	97.78	97.78	97.64	97.71	99.67
	Autoencoder	99.28	99.61	99.6	98.95	99.28	99.97
ResNet-50	Without FR	98.69	98.04	98.06	99.35	98.7	99.82
	SVD	99.74	99.87	99.87	99.61	99.74	100
	Lasso	99.21	99.21	99.22	99.21	99.21	99.98
	Autoencoder	99.15	99.21	99.22	99.08	99.15	99.97

**Table 19 diagnostics-15-03235-t019:** The Computational complexity of the proposed hybrid framework.

Model	Number of Parameters	Number of FLOPs	Feature _Cnn_Extraction Img_Ms	Feature_Hog_ Extraction_ Img_Ms	Feature_Fusion_Ms	Total_Inference_ Time Img_Ms
VGG-19	~144 M parameters	68,900.00 M	24.75206	6.76863	0.00007	31.52964
DenseNet-169	~14 M parameters	11,700.00 M	19.58768	6.77190	0.00010	26.37246
ResNet-50	~26 M parameters	14,491.44 M	18.11749	6.57882	0.00011	24.70996
MobileNet-V2	~3.5 M parameters	1151.26 M	12.57125	6.67308	0.00008	19.25461

**Table 20 diagnostics-15-03235-t020:** Performance comparison with state-of-the-art methods.

Reference	Methodology	Accuracy/AUC	Datasets
Hashem et al. [1]	InceptionResNetV2	89.20%	TPP dataset
Yoo et al. [3]	CNN	AUC of 87%	TPP dataset
Sarıateş and Özbay [23]	DenseNet-201 and RMSProp	98.63%	TPP dataset
Liu et al. [24]	XmasNet (CNNs)	AUC of 84%	TPP dataset
Mehta et al. [25]	PCF (SVM)	ROC of 86%	TPP dataset
Salama and Aly [26]	ResNet-50,VGG-16 and, SVM	98.79%	TPP dataset
Pellicer-Valero et al. [27]	CNN	AUC of 96%	TPP dataset
In Giganti et al. [28]	CNN	AUC of 91%	TPP dataset
The proposed Hybrid Model	MobileNet-V2, HOG, SVD, and SVM	Accuracy of 99.74% and AUC of 100%	TPP dataset

## Data Availability

Our research utilized a publicly available and fully anonymized dataset called “Transverse Plane Prostate Dataset”, which is hosted on Kaggle. This dataset can be found https://www.kaggle.com/datasets/tgprostata/transverse-plane-prostate-dataset (Accessed on 23 November 2025).

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
