# Peer review of "A Multi-Stage Hybrid Learning Model with Advanced Feature Fusion for Enhanced Prostate Cancer Classification"

_diagnostics, 2025, doi:10.3390/diagnostics15243235_

Round 1
Reviewer 1 Report
Comments and Suggestions for Authors
- The contributions mentioned by the author in lines 194–204 are somewhat redundant and should not be listed separately. Experimental methods should not be regarded as contributions. It is recommended to consolidate the innovations into 3–4 distinct points.
- In section 3.1 (TPP Dataset), the author should provide either a citation or an access link for the dataset to facilitate reader access.
- In section 3 (Methods and Datasets), the author describes MobileNet-V2, DenseNet-169, and ResNet-50 but does not specify the innovative aspects of the proposed algorithm. The author should explicitly state their contributions.
- In Table 1 the “Number of epochs” is set to 30. Is this value too low to achieve model convergence? Additionally, the “Optimizer” is incorrectly listed as “Sigmoid”, which is an activation function, not an optimizer. The author should thoroughly proofread the paper to avoid such fundamental errors.
- Figure 8 is rather blurry and its content is difficult to discern. The author should optimize the presentation of this figure.
- The author is recommended to add a table comparing model complexity to demonstrate the impact of the proposed module on the network.
Author Response
Dear Ms. Emma Jiang,
Thank you very much for allowing us to submit a revised version of our manuscript. We thank all the reviewers for their positive feedback and thoughtful comments. The updated version has incorporated their suggestions for improving the manuscript and highlighting its contributions. All the reviewers' concerns have been considered. Those changes are highlighted in the revised paper. We uploaded (a) our point-by-point response to the comments (below) with specific details about the changes that were made in our revised manuscript, (b) an updated manuscript with yellow highlighting indicating changes, and (c) a clean, updated manuscript without highlights.
Best regards,
Dr.Sameh Abd El-Ghany
Response for Reviewer #1 Comments
Comment #1:
The contributions mentioned by the author in lines 194–204 are somewhat redundant and should not be listed separately. Experimental methods should not be regarded as contributions. It is recommended to consolidate the innovations into 3–4 distinct points.
Response:
Thank you for this valuable comment. We appreciate the observation regarding the redundancy in the listed contributions.
Action:
In response to consolidating the contributions in the revised manuscript, we have consolidated the innovations into a clearer set of three distinct points to avoid overlap and improve readability. Additionally, we have removed methodological descriptions from the contribution list, ensuring that only genuine novel contributions are highlighted as recommended. Please see Introduction on page 3.
- Proposed an innovative hybrid architecture that unifies deep semantic representations with handcrafted structural descriptors and applies Truncated SVD to generate a compact, discriminative representation, reducing redundancy while preserving essential diagnostic information in a coordinated multi-stage learning pipeline offering a new direction for feature-level fusion in prostate cancer diagnosis rather than relying on isolated or sequential approaches.
- Designed a feature integration mechanism that adaptively balances the complementary strengths of deep convolutional representations and handcrafted textural cues, achieving superior discriminative power and interpretability compared to conventional deep-only models.
- The proposed framework consistently outperformed state-of-the-art baselines in accuracy, specificity, and sensitivity, establishing a scalable, generalizable, and interpretable foundation for intelligent prostate cancer diagnosis in real-world medical settings.
We have carefully reviewed your comments and made the necessary changes. We genuinely appreciate your dedicated efforts and hope these revisions will meet with your approval.
Comment #2:
In section 3.1 (TPP Dataset), the author should provide either a citation or an access link for the dataset to facilitate reader access.
Response:
We appreciate your observation. We greatly appreciate your observation regarding the citation of the TPP dataset. A formal reference to the TPP dataset was existing at the end of Section 3.1, including a clear citation to ensure that readers can conveniently review and download the dataset. Please see Methods and Datasets on pages 5-6.
The TPP dataset comprises 1,528 prostate MRI images captured in the transverse plane. The images and their classifications originate from the PROSTATEx Dataset and Documentation. The purpose of this dataset is to train a CNN known as Small VGG Net, enabling the classification of new images into clinically significant and clinically non-significant categories for an undergraduate thesis in systems engineering at the Autonomous University of Bucaramanga (UNAB). The images were collected from 64 patients, ensuring each had a single prostate MRI finding for improved training accuracy. The images were converted from DICOM format to JPEG. Subsequently, the im-ages were divided into two groups using a retention method: 30% for the validation group and the remaining 70% for the training group. Consequently, there are two categories (significant and non-significant) further split into training (70%) and validation (30%) groups [20,21]. Figure 1 presents samples from the TPP dataset.
Figure 1. Samples from the TPP dataset.
We have carefully reviewed your comments and made the necessary changes. We genuinely appreciate your dedicated efforts and hope these revisions will meet with your approval.
Comment #3:
In section 3 (Methods and Datasets), the author describes MobileNet-V2, DenseNet-169, and ResNet-50 but does not specify the innovative aspects of the proposed algorithm. The author should explicitly state their contributions.
Response:
Thank you for pointing this out. We appreciate your comments about the contribution of MobileNet-V2, DenseNet-169, and ResNet-50. In the Introduction section, it is noted that MobileNet-V2 (or DenseNet-169 or ResNet-50) was employed to extract deep features for capturing high-level image representations, which were then combined with handcrafted HOG descriptors. DenseNet-169 and ResNet-50 were used to compare their results with those of MobileNet-V2. Please see Introduction on page 3.
This research presents a hybrid method for classifying PCa by combining deep features extracted from MobileNetV2 (or DenseNet-169 or ResNet-50) with handcrafted HOG descriptors. The resulting feature set undergoes dimensionality reduction using Truncated SVD, followed by classification with SVM. HOG is effective in detecting structural patterns, such as lesion boundaries in medical images, by analyzing the distribution of intensity gradients and edge orientations. SVD helps reduce high-dimensional feature sets to lower dimensions while retaining crucial information, which simplifies models, accelerates computations, and decreases memory usage without significantly compromising accuracy. To evaluate performance across various metrics, stratified k-fold cross-validation is employed, ensuring robustness and generalizability. The experiments utilized the TPP dataset for binary classification tasks. To ensure consistent input, the dataset was preprocessed using resizing and normalization techniques, and it was divided into 80% for training and 20% for testing. Comprehensive experiments were conducted with the TPP dataset, including ablation studies to identify the optimal hyperparameters.
We have carefully reviewed your comments and made the necessary changes. We genuinely appreciate your dedicated efforts and hope these revisions will meet with your approval.
Comment #4:
In Table 1 the “Number of epochs” is set to 30. Is this value too low to achieve model convergence? Additionally, the “Optimizer” is incorrectly listed as “Sigmoid”, which is an activation function, not an optimizer. The author should thoroughly proofread the paper to avoid such fundamental errors.
Response:
Thank you for highlighting these important issues. We appreciate your observation to correct the optimizer.
The number of training epochs was initially set to 30 based on preliminary experiments, which indicated that the model reached a stable validation loss before this point.
Action:
In response to correcting the optimizer in the revised manuscript, we have revised the optimizer in Table 1. Please see Model Performance and Insights on page 14.
Table 1: The experiment’s hyperparameter.
|
Parameter |
Value |
|
img_size |
224x224 |
|
Number of epochs |
30 |
|
channels |
3 |
|
Optimizer |
Adam |
|
Initial learning rate |
0.001 |
|
Loss |
binary_crossentropy |
We have carefully reviewed your comments and made the necessary changes. We genuinely appreciate your dedicated efforts and hope these revisions will meet with your approval.
Comment #5:
Figure 8 is rather blurry and its content is difficult to discern. The author should optimize the presentation of this figure.
Response:
Thank you for the valuable suggestion. We appreciate your comment regarding figure 8.
Action:
In response to revising figure 8, we have replaced the original image with a higher-resolution version, enhanced the font size of the labels, and refined the contrast so that all elements are now much clearer. Please see Model Performance and Insights on page 21.
Figure 8: DL models performance comparison across the four ML models
We have carefully reviewed your comments and made the necessary changes. We genuinely appreciate your dedicated efforts and hope these revisions will meet with your approval.
Comment #6:
The author is recommended to add a table comparing model complexity to demonstrate the impact of the proposed module on the network.
Response:
Thank you for this valuable suggestion. We appreciate your comments to add a table comparing model complexity to demonstrate the impact of the proposed module on the network.
Action:
In response to adding a table comparing model complexity, we have added a new table to the revised manuscript summarizing the model complexity of the baseline architecture and the proposed version with the introduced module. The table reports parameter count, FLOPs, and inference time to clearly illustrate the computational impact of the proposed modification. This addition helps clarify that the module provides performance gains with manageable computational overhead. Please see Model Performance and Insights on pages 36-37.
The computational complexity of the proposed hybrid framework, as shown in Table 19, was largely affected by the deep backbone architecture used for feature extraction, the expense of computing handcrafted features, and the combined impact of late-stage feature fusion on overall inference time. The visual backbone networks varied greatly in scale: VGG-19 had about 144 million parameters and required around 68,900 million floating-point operations (FLOPs), making it the most computationally demanding model in comparison. On the other hand, MobileNet was designed compactly with approximately 3.5 million parameters and 1,151 million FLOPs, resulting in the lowest computational load at the backbone level. Intermediate architectures like DenseNet-169 (about 14M parameters, 11,700M FLOPs), ResNet-50 (approximately 26M parameters, 14,491M FLOPs), and DenseNet-169 showed moderate extraction costs while being significantly more efficient than large, single-structured CNNs.
From an inference-time perspective, the overall execution cost of the pipeline—including CNN feature extraction, HOG descriptor calculation, and multi-source feature fusion—demonstrated that total inference time increased with the size of the backbone. The longest total inference time observed was approximately 31.53 ms with the VGG-19 backbone, while the fusion of lightweight CNN features with HOG took as little as around 19.25 ms when using MobileNet. Sequence-based modeling using LSTM after feature fusion added minimal computational overhead (about 0.00007–0.00011 ms), indicating that the main complexity arose from backbone-driven feature computation and the high dimensionality produced during fusion.
Table 19: The Computational complexity of the proposed hybrid framework.
|
Model |
Number of parameters |
Number of FLOPs |
Feature _cnn_Extraction img_ms |
Feature _hog_ Extraction_ img_ms |
Feature _fusion_ ms |
total_inference_ Time img_ms |
|
VGG-19 |
~ 144 M parameters |
68,900.00 M |
24.75206 |
6.76863 |
0.00007 |
31.52964 |
|
DenseNet-169 |
~ 14 M parameters |
11,700.00 M |
19.58768 |
6.77190 |
0.00010 |
26.37246 |
|
ResNet-50 |
~ 26 M parameters |
14,491.44 M |
18.11749 |
6.57882 |
0.00011 |
24.70996 |
|
MobileNet |
~ 3.5 M parameters |
1,151.26 M |
12.57125 |
6.67308 |
0.00008 |
19.25461 |
We have carefully reviewed your comments and made the necessary changes. We genuinely appreciate your dedicated efforts and hope these revisions will meet with your approval.

Reviewer 2 Report
Comments and Suggestions for Authors
This study meets modern requirements in the field of optimizing the diagnosis of the tumor process.
Automated machine diagnostic process of tumor growth will significantly improve the work of oncologists and help to more quickly assimilate this work with artificial intelligence.
At the same time, there are a number of questions for the authors
1 Was a study of correlations with the results of histological examination conducted
2 Were the results of machine analysis compared with the data of biopsy puncture examination of the prostate
3 Did your studies correlate with the Gleason grading of prostate cancer
Author Response
Dear Ms. Emma Jiang,
Thank you very much for allowing us to submit a revised version of our manuscript. We thank all the reviewers for their positive feedback and thoughtful comments. The updated version has incorporated their suggestions for improving the manuscript and highlighting its contributions. All the reviewers' concerns have been considered. Those changes are highlighted in the revised paper. We uploaded (a) our point-by-point response to the comments (below) with specific details about the changes that were made in our revised manuscript, (b) an updated manuscript with yellow highlighting indicating changes, and (c) a clean, updated manuscript without highlights.
Best regards,
Dr.Sameh Abd El-Ghany
Response for Reviewer #2 Comments
Comment #1:
Was a study of correlations with the results of histological examination conducted?
Response:
Thank you for this valuable question. We appreciate your question about correlations with the results of histological examination.
A study evaluating correlations with histological examination results was not conducted in this research. The experiments were limited to imaging-based labels from the dataset, and therefore the statistical relationship between model predictions and tissue-level clinical findings remained unverified. Establishing such correlations is recommended in AI diagnostic studies to support clinical validity and avoid spurious associations.
This gap was identified as a key direction for future work, where planned studies included collecting paired histological outcomes and performing formal correlation analysis to measure consistency between model outputs and histopathological findings.
We have carefully reviewed your comments and made the necessary changes. We genuinely appreciate your dedicated efforts and hope these revisions will meet with your approval.
Comment #2:
Were the results of machine analysis compared with the data of biopsy puncture examination of the prostate?
Response:
Thank you for this valuable question. We appreciate your question about the comparison between the results of machine analysis with the data of biopsy puncture examination of the prostate.
The results were not compared with prostate biopsy puncture data. The study was performed on the Transverse Plane Prostate (TPP) dataset, which contained only MRI images and did not include any biopsy, histopathology, or clinical puncture examination records. Because the dataset was strictly image-based, there was no available ground-truth for biopsy puncture examination of the prostate, making such a comparison infeasible
We have carefully reviewed your comments and made the necessary changes. We genuinely appreciate your dedicated efforts and hope these revisions will meet with your approval.
Comment #3:
Did your studies correlate with the Gleason grading of prostate cancer.
Response:
Thank you for this valuable question. We appreciate your question about the correlation with the Gleason grading of prostate cancer.
The research did not include a correlation with Gleason grading. The dataset contained a mix of diagnostically clear and ambiguous MRI cases, where some images strongly expressed pathological cues while others showed less distinctive or statistically non-significant patterns. The model was designed solely to perform binary detection of prostate cancer rather than severity stratification. Future work was planned to incorporate Gleason grading by evaluating class-wise separability, labeling images according to grade-level clinical annotations, and extending the classifier to support multi-stage cancer severity prediction.
We have carefully reviewed your comments and made the necessary changes. We genuinely appreciate your dedicated efforts and hope these revisions will meet with your approval.
Reviewer 3 Report
Comments and Suggestions for Authors
First of all, I would like to thank the authors of this article for their efforts. This article is mainly about a multi-stage hybrid learning model with advanced feature fusion for enhanced prostate cancer classification. Although the overall quality of this article is good, especially its detailed methods, the following points are recommended to further improve its quality.
1. In the abstract, please add the result section and mention all important metrics with sufficient statistics. It is also recommended to briefly mention the dataset used.
2. The introduction of your study is too long, it is recommended to summarize it to approximately two pages.
3. In the last paragraph of the introduction, please clearly mention the gaps in current knowledge, your objectives, and the novelty of your study. It is also recommended to cite recent articles to support your statements.
4. Please ensure that Section 2 focuses only on the subject of your study. Some of the mentioned items are duplicates with the introduction, which should be deleted.
5. When citing specific studies, please use the last name of the first author (e.g., instead of Liu, S. et al., write Liu et al.).
6. In the method section, please mention how you dealt with overfitting bias?
7. Another important point in method section is about how you assessed the quality of included images? Did you use augmentation methods? Also, please mention your detailed inclusion and exclusion criteria.
8. In the result section, it is also recommended to mention some baseline characteristics of your dataset in details.
9. In the discussion section, it is also recommended to discuss your study limitations and your recommendations for future research. Please expand this section and use more references.
10. If you used AI-gen tools at any stage of preparing this article (including writing), please disclose.
11. As another point, please re-organize your manuscript structure based on CLAIMS reporting checklist. Please, if possible, complete and revise your study according to the mentioned items in the CLAIMS reporting checklist. Please also submit the completed version of CLAIMS based on your revised manuscript in revision stage.
12. In the figure legends, please briefly mention any tools you used to generate and create these figures (if these figures were generated by artificial intelligence, please also mention them).
13. It is recommended that some general items of the Methods be considered as supplementary material to the main manuscript. It is also necessary to mention some formulas used in the method in this file.
14. There are some sentences in the method that cannot be left without reference. Please check that all parts that refer to previous studies or previous methods have appropriate references.
15. Please separate the results from the method in section 4. For example, 4.1 4.2 and the initial parts of 4.3 are related to the method.
Author Response
Dear Ms. Emma Jiang,
Thank you very much for allowing us to submit a revised version of our manuscript. We thank all the reviewers for their positive feedback and thoughtful comments. The updated version has incorporated their suggestions for improving the manuscript and highlighting its contributions. All the reviewers' concerns have been considered. Those changes are highlighted in the revised paper. We uploaded (a) our point-by-point response to the comments (below) with specific details about the changes that were made in our revised manuscript, (b) an updated manuscript with yellow highlighting indicating changes, and (c) a clean, updated manuscript without highlights.
Best regards,
Dr.Sameh Abd El-Ghany
Response for Reviewer #3 Comments
Comment #1:
In the abstract, please add the result section and mention all important metrics with sufficient statistics. It is also recommended to briefly mention the dataset used.
Response:
Thank you for this valuable suggestion. We appreciate the suggestion regarding the mention of the results and the dataset in the abstract.
Action:
In response to mention results and the dataset in the abstract, we have revised the abstract to include a concise results section that reports all key performance metrics with appropriate statistical detail. Additionally, we now provide the name of the dataset used in the study to give readers clearer context regarding the experimental setup and data characteristics. Please see Abstract on page 1.
Abstract: Background: Cancer poses a significant health risk to humans, with prostate cancer (PCa) being the second most common and deadly form among men, following lung cancer. Each year, it affects over a million individuals and presents substantial diagnostic challenges due to variations in tissue appearance and imaging quality. In recent decades, various techniques utilizing Magnetic Resonance Imaging (MRI) have been developed for identifying and classifying PCa. Accurate classification in MRI typically requires the integration of complementary feature types, such as deep semantic representations from Convolutional Neural Networks (CNNs) and handcrafted descriptors like Histogram of Oriented Gradients (HOG). Therefore, a more robust and discriminative feature integration strategy is crucial for enhancing computer-aided diagnosis performance. Objectives: This study aims to develop a multi-stage hybrid learning model that combines deep and handcrafted features, investigates various feature reduction and classification techniques, and improves diagnostic accuracy for prostate cancer using magnetic resonance imaging. Methods: The proposed framework integrates deep features extracted from convolutional architectures with handcrafted texture descriptors to capture both semantic and structural information. Multiple dimensionality reduction methods, including singular value decomposition (SVD), were evaluated to optimize the fused feature space. Several machine learning (ML) classifiers were benchmarked to identify the most effective diagnostic configuration. The overall framework was validated using k-fold cross-validation to ensure reliability and minimize evaluation bias. Conclusions: Experimental results on the Transverse Plane Prostate (TPP) dataset for binary classification tasks showed that the hybrid model significantly outperformed individual deep or handcrafted approaches, achieving superior accuracy of 99.74%, specificity of 99.87%, precision of 99.87%, sensitivity of 99.61%, and F1-score of 99.74%. By combining complementary feature extraction, dimensionality reduction, and optimized classification, the proposed model offers a reliable and generalizable solution for prostate cancer diagnosis and demonstrates strong potential for integration into intelligent clinical decision-support systems.
We have carefully reviewed your comments and made the necessary changes. We genuinely appreciate your dedicated efforts and hope these revisions will meet with your approval.
Comment #2:
The introduction of your study is too long, it is recommended to summarize it to approximately two pages.
Response:
Thank you for your observation, we appreciate your observation regarding the length of the introduction.
Action:
In accordance with the suggestion, the introduction has been carefully revised and condensed to improve clarity and focus. The updated version is now approximately two pages, emphasizing the essential background, the motivation of problem, and the study objectives while removing non-critical details. This revision enhances readability and aligns the section with standard manuscript guidelines. Please see Introduction on pages 1-3.
- Introduction
The prostate functions as a muscle-driven mechanical switch between urine flow and ejaculation, serving as an auxiliary gland in the male reproductive system [1]. Located within the pelvis and encircling the urethra, this small gland resembles a walnut in shape. Its primary role is to produce prostate fluid, which combines with sperm from the testicles to form semen [2]. As men age, the prostate tends to enlarge, which can impede the urinary flow between the bladder and urethra [1]. Figure 1 shows the physiology of a human prostate.
According to the World Health Organization (WHO), PCa ranks as the second most prevalent cancer diagnosed in men and is a leading cause of cancer-related deaths globally. In 2020, there were 1,414,259 new cases—accounting for 7.3% of all new cancer cases—alongside 375,304 deaths, representing 3.8% of total cancer fatalities worldwide. Projections indicate these figures will climb to 2,430,000 new cases and 740,000 deaths by 2040 [3]. In the United States, PCa was the third leading cause of cancer death among men in 2017, with approximately 161,360 new cases, making up 19% of all new cancer diagnoses, and resulting in 26,730 deaths, which constituted 8% of all cancer deaths [4]. In Egypt, PCa is also prevalent among men, with 4,767 new cases (3.5% of all new cancer cases) and 2,227 deaths (2.5% of total cancer deaths). By 2040, these figures are expected to double, reaching 9,610 new cases and 4,980 deaths [5]. These statistics highlight a significant increase in the morbidity and mortality rates of PCa, indicating it is one of the fastest-growing malignancies among men. Therefore, early detection and diagnosis of PCa are essential for enhancing patient care and improving survival rates.
Many PCa cases develop very slowly, resulting in minimal issues that don't require treatment. However, in some instances, PCa can progress rapidly and invade nearby tissues and organs [6]. The most prevalent form of PCa is adenocarcinoma. Other var-iants include transitional cell carcinomas, neuroendocrine tumors, sarcomas, small cell carcinomas, and squamous cell carcinomas [7].
The manifestation of PCa symptoms varies according to the disease stage—early, advanced, or recurrent. In its initial phase, PCa frequently shows no symptoms and may progress slowly, often needing little to no treatment. As the condition advances, various symptoms may arise. Common issues reported include: (1) difficulty urinating, (2) re-duced strength of the urine stream, (3) presence of blood in urine, (4) painful ejaculation, (5) blood in semen, and (6) erectile dysfunction. In later or metastatic stages, patients might experience systemic symptoms such as (7) bone pain, (8) fatigue, (9) unexplained weight loss, (10) numbness in the legs or feet due to spinal cord compression, and (11) jaundice if the liver is affected [8,9].
The risk factors associated with PCa are influenced by an individual's lifestyle, age, and family background. Factors that elevate the likelihood of developing PCa include (1) obesity, (2) advancing age (particularly after 50), (3) a family history of the disease, (4) ethnicity (notably, Black men are at a higher risk for being diagnosed with PCa), and (5) genetic alterations or changes in cellular DNA [10].
Multiparametric MRI (mpMRI) has emerged as a crucial tool for detecting and as-sessing PCa. MpMRI integrates anatomical imaging with functional sequences, in-cluding diffusion-weighted imaging (DWI) and dynamic contrast-enhanced (DCE) imaging, enhancing the differentiation between benign tissue and clinically significant tumors. This method not only improves cancer localization and staging but also mini-mizes unnecessary biopsies by directing targeted sampling of suspicious regions. Ad-ditionally, MRI has been demonstrated to reduce the overdiagnosis of indolent cancers while enhancing the identification of aggressive disease, thus facilitating more person-alized treatment options. However, despite its clear benefits in detecting PCa, MRI also has several drawbacks and limitations [1]: MRI scans tend to be much costlier than ul-trasound or biopsy and may not be easily accessible, particularly in areas with limited resources. MpMRI involves longer scanning durations (30–45 minutes), which restricts the number of patients that can be accommodated compared to alternative imaging methods. The accuracy of diagnoses relies significantly on the skills and experience of the radiologist. Non-cancerous conditions (such as prostatitis and benign prostatic hyperplasia) can resemble cancer, resulting in false-positive results.
Pca can be managed using several approaches, depending on its stage and pro-gression. Localized cases may be treated with surgery (prostatectomy), radiotherapy (either brachytherapy or external beam), active surveillance, watchful waiting, or focal therapy. When the disease is confined to a specific area, targeted treatments such as focal therapy are used, while cryotherapy may be selected for patients unable to un-dergo surgery or radiation. If the cancer spreads beyond the prostate, systemic treat-ments become necessary [11,12].
DL models, particularly Convolutional Neural Networks (CNNs), have shown remarkable proficiency in tasks such as image recognition, segmentation, and classifi-cation across multiple fields, including medical imaging. Their capacity to autonomously learn complex features from intricate datasets, eliminating the necessity for manual feature extraction, renders them highly effective for analyzing the extensive and complicated data produced by MRI scans [13].
This research presents a hybrid method for classifying PCa by combining deep features extracted from MobileNetV2 with handcrafted HOG descriptors. The resulting feature set undergoes dimensionality reduction using Truncated SVD, fol-lowed by classification with SVM. HOG is effective in detecting structural patterns, such as lesion boundaries in medical images, by analyzing the distribution of intensity gradients and edge orientations. SVD helps reduce high-dimensional feature sets to lower dimensions while retaining crucial information, which simplifies models, accelerates computations, and decreases memory usage without significantly compromising ac-curacy. To evaluate performance across various metrics, stratified k-fold cross-validation is employed, ensuring robustness and generalizability. The experiments utilized the TPP dataset for binary classification tasks. To ensure consistent input, the dataset was preprocessed using resizing and normalization techniques, and it was divided into 80% for training and 20% for testing. Comprehensive experiments were conducted with the TPP dataset, including ablation studies to identify the optimal hyperparameters. A summary of our research our contributions is outlined below:
- Proposed an innovative hybrid architecture that unifies deep semantic representations with handcrafted structural descriptors and applies Truncated SVD to generate a compact, discriminative representation, reducing redundancy while preserving essential diagnostic information in a coordinated multi-stage learning pipeline offering a new direction for feature-level fusion in prostate cancer diagnosis rather than relying on isolated or sequential approaches.
- Designed a feature integration mechanism that adaptively balances the comple-mentary strengths of deep convolutional representations and handcrafted textural cues, achieving superior discriminative power and interpretability compared to conventional deep-only models.
- The proposed framework consistently outperformed state-of-the-art baselines in accuracy, specificity, and sensitivity, establishing a scalable, generalizable, and in-terpretable foundation for intelligent prostate cancer diagnosis in real-world medical settings.
1.1. Paper Structure
The remainder of this paper is structured as follows: Section 2 reviews the current literature on PCa using the TPP dataset. Section 3 details the TPP dataset and outlines our methodology. Section 4 presents the experimental results of the proposed hybrid model. Lastly, Section 5 concludes with a summary of our findings.
We have carefully reviewed your comments and made the necessary changes. We genuinely appreciate your dedicated efforts and hope these revisions will meet with your approval.
Comment #3:
In the last paragraph of the introduction, please clearly mention the gaps in current knowledge, your objectives, and the novelty of your study. It is also recommended to cite recent articles to support your statements.
Response:
Thank you for this valuable suggestion. We appreciate your comments about mention clearly the gaps in current knowledge, our objectives, and the novelty of our study.
The final paragraph of the Literature review (Section 2) explicitly highlights the existing gaps in current knowledge, clearly states the objectives of the study, and articulates the novelty of our proposed approach. Please see Literature Review on page 5.
The limitations of the earlier studies are outlined as follows:
1.In the previous research, the authors did not utilize HOG descriptors or SVD. However, we developed an innovative approach that integrates deep features from MobileNetV2 with handcrafted HOG descriptors to effectively capture both se-mantic and structural patterns in PCa images. Additionally, we employed SVD to compress the combined feature set, which minimized redundancy, enhanced computational speed, and reduced memory usage while retaining crucial diagnostic information.
2.The researchers in the previously referenced study did not conduct an ablation study. In contrast, we performed this analysis to evaluate the impact of each component or feature of our proposed model on its performance, systematically modifying or excluding elements and analyzing their effects.
We have carefully reviewed your comments and made the necessary changes. We genuinely appreciate your dedicated efforts and hope these revisions will meet with your approval.
Comment #4:
Please ensure that Section 2 focuses only on the subject of your study. Some of the mentioned items are duplicates with the introduction, which should be deleted.
Response:
Thank you for the valuable feedback. We appreciate your feedback about removing duplication.
In Section 2, we reviewed the literature on PCa detection, highlighted the current knowledge gaps, clearly stated the study's objectives, and explained the novelty of our proposed approach. If we added the literature to the Introduction, it will be too long. Please see Literature Review on pages 4 and 5
- Literature Review
The diagnosis of PCa is a significant focus in the field of medical image analysis. Many studies tackle this problem from diverse perspectives. For example, Hashem, H. et al. [1] proposed an MRI-based technique to enhance diagnostic accuracy. This method involved two primary phases. Initially, the MRI images underwent preprocessing to optimize them for the detection phase. Following this, PCa detection was conducted using the InceptionResNetV2 model. The InceptionResNetV2 model achieved an average accuracy of 89.20% and an area under the curve (AUC) of 93.6%.
Yoo, S. et al. [13] designed and executed an automated pipeline utilizing CNN to identify clinically significant PCa from axial DWI for individual patients. To evaluate the effectiveness of the proposed pipeline, a testing subset of 108 patients (from the original 427) was reserved and excluded from the training process. The pipeline demonstrated an (AUC of 0.87 (95% Confidence Interval: 0.84–0.90) at the slice level and 0.84 (95% CI: 0.76–0.91) at the patient level.
In Sarıateş, M. and Özbay, E. [14], to enhance the classification accuracy for PCa diagnosis, transfer learning techniques and fine-tuning processes were utilized. A two-class dataset, comprising MR images of PCa labeled as 'significant' and 'not-significant', was classified using AlexNet, DenseNet-201, GoogleNet, and VGG-16 models through a feature extraction strategy, yielding accuracy rates of 71.40%, 72.05%, 65%, and 80.13%, respectively. To further improve these results, pre-trained transfer learning models were implemented, achieving accuracy rates of 89.74%, 94.32%, 85.59%, and 91.05%, respectively. Additionally, a validation accuracy of 98.10% was attained using the cross-validation method with the DenseNet-201 model. The DenseNet-201 model reached the highest accuracy of 98.63% when combined with the RMSProp optimization technique.
Liu, S. et al. [15] proposed a DL framework named XmasNet, which utilized CNNs, for classifying PCa lesions using 3D multiparametric MRI data from the PROSTATEx challenge. The XmasNet model underwent end-to-end training, with data augmentation achieved through 3D rotations and slicing to effectively capture the 3D characteristics of the lesions. In the testing phase, XmasNet outperformed 69 different methods from 33 competing teams, earning the second highest AUC score of 84% in the PROSTATEx challenge.
Mehta, P. et al. [16] presented a new patient-level classification framework called PCF, which is trained exclusively on patient-level labels. In the PCF approach, features are extracted from three-dimensional mpMRI and related parameter maps using CNNs. These features were then integrated with clinical data through a multi-classifier SVM. The result of the PCF framework was a probability score indicating whether a patient has clinically significant PCa based on the Gleason score.
PCF achieved average area under the receiver operating characteristic (ROC) curves of 0.79 and 0.86 for the PICTURE and PROSTATEx datasets, respectively, using five-fold cross-validation.
Salama, W.M. and Aly, M.H. [17] examined four distinct strategies for the classification task. Both ResNet-50 and VGG-16 architectures were employed and re-trained to analyze the diffusion-weighted DWI database, aiming to determine the presence of PCa. To address the challenge of insufficient labeled data and enhance system efficiency, transfer learning and data augmentation techniques were implemented for both ResNet50 and VGG-16. The final fully connected layer was substituted with a SVM classifier to improve accuracy. Additionally, both transfer learning and data augmentation processes were applied to the SVM to bolster the performance of our framework. A k-fold cross-validation method was also utilized to evaluate model performance. Authors utilized end-to-end fully CNNs, eliminating the need for preprocessing or post-processing steps. The technique combining ResNet-50 with SVM yields the highest performance, achieving an accuracy of 98.79%, ana AUC of 98.91%, sensitivity of 98.43%, precision of 97.99%, an F1 score of 95.92%, and a computational time of 2.345 seconds.
Pellicer-Valero, O.J. et al. [18] introduced a fully automated system utilizing DL to localize, segment, and estimate the Gleason grade group (GGG) of PCa lesions from mpMRI scans. The system was trained and validated on distinct datasets: ProstateX and the Valencian Oncology Institute Foundation. In the testing phase, it achieved impressive lesion-level performance metrics for the GGG 2 significance criterion, with an AUC, sensitivity, and specificity of 0.96, 1.00, and 0.79, respectively, for the ProstateX dataset, and 0.95, 1.00, and 0.80 for the IVO dataset. At the patient level, the results were 0.87, 1.00, and 0.375 for ProstateX, and 0.91, 1.00, and 0.762 for IVO. Additionally, in the online ProstateX grand challenge, the model achieved an AUC of 0.85 (increased to 0.87 when trained solely on ProstateX data), matching the performance of the original challenge winner. For comparison with expert evaluations, the sensitivity and specificity of the IVO radiologist’s PI-RADS 4 were 0.88 and 0.56 at the lesion level, and 0.85 and 0.58 at the patient level.
In Giganti, F. et al. [19], a DL computer-aided detection (CAD) medical device marked with Conformité Européenne (CE) (referred to as Pi) was developed to identify Gleason Grade Group (GG) ≥ 2 cancer using historical data from the PROSTATEx dataset and five hospitals in the UK, involving 793 patients. The prevalence of GG ≥ 2 in the validation dataset was found to be 31%. When assessed on a per-patient basis, Pi demonstrated non-inferiority to the radiologists (with a performance difference of 10% deemed acceptable), achieving an AUC of 0.91 compared to 0.95 for the radiologists. At the established risk threshold of 3.5, the AI system's sensitivity was 95% and specificity was 67%. In contrast, radiologists using Prostate Imaging-Reporting and Data Systems/Likert scores of ≥ 3 detected GG ≥ 2 with a sensitivity of 99% and specificity of 73%. AI performance was consistent across different sites (AUC ≥ 0.83) at the patient level, regardless of the scanner's age and field strength.
The limitations of the earlier studies are outlined as follows:
- In the previous research, the authors did not utilize HOG descriptors or SVD. However, we developed an innovative approach that integrates deep features from MobileNetV2 with handcrafted HOG descriptors to effectively capture both semantic and structural patterns in PCa images. Additionally, we employed SVD to compress the combined feature set, which minimized redundancy, enhanced computational speed, and reduced memory usage while retaining crucial diagnostic information.
- The researchers in the previously referenced study did not conduct an ablation study. In contrast, we performed this analysis to evaluate the impact of each component or feature of our proposed model on its performance, systematically modifying or excluding elements and analyzing their effects.
We have carefully reviewed your comments and made the necessary changes. We genuinely appreciate your dedicated efforts and hope these revisions will meet with your approval.
Comment #5:
When citing specific studies, please use the last name of the first author (e.g., instead of Liu, S. et al., write Liu et al.).
Response:
Thank you for the helpful suggestion. We appreciate your helpful suggestion regarding citation.
We formatted the references according to the citation style of MDPI journals.
We have carefully reviewed your comments and made the necessary changes. We genuinely appreciate your dedicated efforts and hope these revisions will meet with your approval.
Comment #6:
In the method section, please mention how you dealt with overfitting bias?
Response:
Thank you for highlighting the need to clarify how overfitting was addressed. We appreciate your comments about overfitting. To reduce overfitting and improve generalization, we implemented several strategies in our pipeline. First, we employed Stratified K-Fold cross-validation, which ensures that each fold preserves the class distribution, providing a robust estimate of model performance across unseen data. Second, for CNN-based feature extraction, we froze the weights of the pretrained CNN models' layers and only used the output of the Global Average Pooling layer as fixed features, preventing the network from overfitting on the small dataset. Third, when combining features (CNN + HOG), we applied Standard Scaling to normalize feature distributions, which stabilizes classifier training. Finally, for the machine learning classifiers (XGBoost, CatBoost, SVM, etc.), we used default regularization parameters and limited model complexity (e.g., max depth, learning rate) to further reduce the risk of overfitting
We have carefully reviewed your comments and made the necessary changes. We genuinely appreciate your dedicated efforts and hope these revisions will meet with your approval.
Comment #7:
Another important point in method section is about how you assessed the quality of included images? Did you use augmentation methods? Also, please mention your detailed inclusion and exclusion criteria.
Response:
Thank you for highlighting the need to clarify the assessment of the quality of images included. We appreciate your comments about the image’s quality.
In our study, we used the Transverse Plane Prostate Dataset that is part of PROSTATEx dataset, which provides expert-validated lesion annotations. Image quality was ensured by relying on these radiologist-reviewed coordinates and metadata, confirming accurate lesion localization and anatomical correctness. Inclusion criteria consisted of MRI exams with complete multi-parametric sequences and validated lesion labels, while cases with missing sequences or poor image registration were excluded.
We have carefully reviewed your comments and made the necessary changes. We genuinely appreciate your dedicated efforts and hope these revisions will meet with your approval.
Comment #8:
In the result section, it is also recommended to mention some baseline characteristics of your dataset in details.
Response:
Thank you for the constructive feedback. We appreciate your feedback about baseline characteristics of the used dataset. The baseline characteristics of the used dataset were described in subsection 3.1. This will help contextualize the results and strengthen the manuscript’s transparency. Please see Methods and Datasets on pages 5 and 6.
The TPP dataset comprises 1,528 prostate MRI images captured in the transverse plane. The images and their classifications originate from the PROSTATEx Dataset and Documentation. The purpose of this dataset is to train a CNN known as Small VGG Net, enabling the classification of new images into clinically significant and clinically non-significant categories for an undergraduate thesis in systems engineering at the Autonomous University of Bucaramanga (UNAB). The images were collected from 64 patients, ensuring each had a single prostate MRI finding for improved training accuracy. The images were converted from DICOM format to JPEG. Subsequently, the images were divided into two groups using a retention method: 30% for the validation group and the remaining 70% for the training group. Consequently, there are two categories (significant and non-significant) further split into training (70%) and validation (30%) groups [20,21]. Figure 1 presents samples from the TPP dataset.
We have carefully reviewed your comments and made the necessary changes. We genuinely appreciate your dedicated efforts and hope these revisions will meet with your approval.
Comment #9:
In the discussion section, it is also recommended to discuss your study limitations and your recommendations for future research. Please expand this section and use more references.
Response:
Thank you for this valuable feedback. We appreciate your feedback about our study limitations and our recommendations for future research.
In subsection 4.6, we discussed the limitations and future work of our study. If we added them to subsection 4.5, it will be too long. Please see Model Performance and Insights on pages 40-41.
Limitations and Future Work
The proposed hybrid model demonstrated superior generalization and discriminative power, making it the most effective approach among those compared on the TPP dataset. However, it has several limitations. The RF classifier performed significantly worse, indicating that simpler models are not well-suited for the high-dimensional deep features generated by MobileNet-V2. While the ensemble methods (XGB and CatBoost) showed strong performance, they were still outperformed by the SVM, suggesting that the boosting techniques may not have fully captured the nonlinear boundaries within the feature space.
Another limitation is the lack of cross-dataset validation and real-world clinical testing, which restricts the assessment of robustness. Additionally, the computational complexity and training time were not thoroughly analyzed, which could be important for practical deployment in resource-constrained environments.
Future research should focus on addressing these limitations by incorporating domain adaptation techniques or transferring learning from larger medical datasets to enhance performance. Integrating explainable AI (XAI) methods, such as Grad-CAM or SHAP, could improve model transparency and interpretability for clinicians. Moreover, real-time deployment and validation in clinical workflows should be pursued to evaluate feasibility, usability, and the impact on diagnostic efficiency in real-world settings.
We have carefully reviewed your comments and made the necessary changes. We genuinely appreciate your dedicated efforts and hope these revisions will meet with your approval.
Comment #10:
If you used AI-gen tools at any stage of preparing this article (including writing), please disclose.
Response:
Thank you for highlighting this out. We appreciate your comments about article writing. We confirm that no AI-generated tools were used at any stage of preparing this manuscript, including writing, analysis, or figure generation. All content was developed and refined solely by the authors.
We have carefully reviewed your comments and made the necessary changes. We genuinely appreciate your dedicated efforts and hope these revisions will meet with your approval.
Comment #11:
As another point, please re-organize your manuscript structure based on CLAIMS reporting checklist. Please, if possible, complete and revise your study according to the items mentioned in the CLAIMS reporting checklist. Please also submit the completed version of CLAIMS based on your revised manuscript in revision stage.
Response:
We appreciate you for the valuable recommendation. We have carefully reorganized the manuscript to align with the CLAIMS reporting checklist. All relevant items have been addressed and incorporated into the manuscript to ensure compliance with the checklist’s standards (Title, Abstract, Introduction, Methods, Evaluation (Training and Testing), Results, and Discussion).
We have carefully reviewed your comments and made the necessary changes. We genuinely appreciate your dedicated efforts and hope these revisions will meet with your approval.
Comment #12:
In the figure legends, please briefly mention any tools you used to generate and create these figures (if these figures were generated by artificial intelligence, please also mention them).
Response:
We appreciate you for the suggestion. The figures related to the results were created from python language and the others were developed and refined solely by the authors using MS Excel and word.
We have carefully reviewed your comments and made the necessary changes. We genuinely appreciate your dedicated efforts and hope these revisions will meet with your approval.
Comment #13:
It is recommended that some general items of the Methods be considered as supplementary material to the main manuscript. It is also necessary to mention some formulas used in the method in this file.
Response:
We appreciate your suggestion. In response, we have included the methodological details directly in the manuscript, as there is no supplementary material. Additionally, we have incorporated the five key formulas used in our methods within the manuscript, since the supplementary files will not be published alongside
We have carefully reviewed your comments and made the necessary changes. We genuinely appreciate your dedicated efforts and hope these revisions will meet with your approval.
Comment #14:
There are some sentences in the method that cannot be left without reference. Please check that all parts that refer to previous studies or previous methods have appropriate references.
Response:
We appreciate you for highlighting this point.
Action:
We have carefully reviewed the Methods and Datasets section and ensured that all statements referring to previous studies, established techniques, or prior methodologies now include appropriate citations. References have been added to support these statements, ensuring proper acknowledgment of prior work and enhancing the rigor of the manuscript.
We have carefully reviewed your comments and made the necessary changes. We genuinely appreciate your dedicated efforts and hope these revisions will meet with your approval.
Comment #15:
Please separate the results from the method in section 4. For example, 4.1 4.2 and the initial parts of 4.3 are related to the method.
Response:
Thanks for your suggestion. We appreciate your suggestion regarding the Method and Results sections.
Action:
In response to moving subsections 4.1 and 4.2, we have been revised the manuscript to clearly separate the Methods and Results in Section 4. Subsections 4.1 and 4.2, which previously included methodological details, have been moved to the Methods section, ensuring that Section 4 now exclusively presents the results. This change improves the clarity and logical flow of the manuscript. Please see Methods and Datasets on pages 13 and 14.
3.2.9 Evaluated Performance Metrics
The effectiveness of the proposed hybrid DL model was evaluated using the formulas presented in Equations (1) through (7).
True Positive (TP) refers to instances where the model accurately identifies positive cases, meaning the prediction aligns with the actual positive data in the dataset. True Negative (TN) signifies that the model successfully predicts negative cases, and this prediction is also accurate in the dataset. False Positive (FP) occurs when the model mistakenly identifies a positive case, whereas the actual outcome is negative (indicating no tumor) in the dataset. False Negative (FN) happens when the model incorrectly predicts a negative case, even though the actual result is positive (indicating the presence of a tumor) in the dataset. The total number of patients with tumors is represented by the sum of TP and FN.
The AUC-ROC is a widely used performance metric for evaluating binary classification models. It measures the model’s ability to discriminate between positive and negative classes across all possible classification thresholds. The ROC curve plots the true positive rate (sensitivity) against the false positive rate (1 – specificity), and the AUC quantifies the overall separability achieved by the model. An AUC value of 1.0 indicates perfect classification, whereas a value of 0.5 suggests performance no better than random chance. Higher AUC values signify better model performance in distinguishing between classes, making it particularly useful in medical imaging and diagnostic tasks, such as detecting PCa from mpMRI images, where accurate discrimination between disease and non-disease cases is critical.
3.2.10 Experiments Setup
In this research, we conducted four experiments to assess the classification performance of the proposed hybrid model using the TPP dataset. These experiments were executed in a Kaggle environment featuring an Intel Core i7 (11th Gen) processor and 16 GB of RAM. For the implementation, we employed Python 3 alongside TensorFlow, a DL framework created by Google. The specific hyperparameters used in the experiment include learning rate, batch size, number of epochs, and type of optimizer outlined in Table 1.
We have carefully reviewed your comments and made the necessary changes. We genuinely appreciate your dedicated efforts and hope these revisions will meet with your approval.
Round 2
Reviewer 1 Report
Comments and Suggestions for Authors
The authors have fully addressed all my concerns; I recommend acceptance of the paper.
Author Response
Comments and Suggestions for Authors
The authors have fully addressed all my concerns; I recommend acceptance of the paper.
Replay,
Thank you very much for your effort and valuable comments, which played a vital role in enhancing the manuscript.
Reviewer 3 Report
Comments and Suggestions for Authors
First of all, I would like to thank the authors for their efforts in addressing the peer reviewers’ comments. Although the overall quality has improved, many comments were left unanswered or not properly answered. For example, comment 1 was about adding the Results section to the abstract (after the Methods, before the Conclusion), which incorrectly added the results to the Conclusion, which is wrong. Or in the case of comment 4, the authors continued to refer to things that were not necessary. In the case of comment 5, the authors mistakenly reflected exactly the state of the References section in the text, which is against the formal language of academic writing. Please change it as in the example provided in the previous revision. The changes following comments 6 and 7 have not been made to the text. Similarly, comment 12 has not been made, and the completed checklist has not been submitted as an appendix.
Author Response
Dear Ms. Emma Jiang,
Thank you very much for allowing us to submit a revised version of our manuscript. We thank all the reviewers for their positive feedback and thoughtful comments. The updated version has incorporated their suggestions for improving the manuscript and highlighting its contributions. All the reviewers' concerns have been considered. Those changes are highlighted in the revised paper. We uploaded (a) our point-by-point response to the comments (below) with specific details about the changes that were made in our revised manuscript, (b) an updated manuscript with yellow highlighting indicating changes, and (c) a clean, updated manuscript without highlights.
Best regards,
Dr. Sameh Abd El-Ghany
Response for Reviewer #3 Comments
Comment #1:
In the abstract, please add the result section and mention all important metrics with sufficient statistics. It is also recommended to briefly mention the dataset used.
Response:
Thank you for this valuable suggestion. We appreciate the suggestion regarding the mention of the results and the dataset in the abstract.
Action:
In response to adding result section and the dataset in the abstract, we have revised the abstract to include a result section that reports all key performance metrics with appropriate statistical detail. Additionally, we now provide the name of the dataset used in the study to give readers clearer context regarding the experimental setup and data characteristics. Please see Abstract on page 1.
Abstract: Background: Cancer poses a significant health risk to humans, with prostate cancer (PCa) being the second most common and deadly form among men, following lung cancer. Each year, it affects over a million individuals and presents substantial diagnostic challenges due to variations in tissue appearance and imaging quality. In recent decades, various techniques utilizing Magnetic Resonance Imaging (MRI) have been developed for identifying and classifying PCa. Accurate classification in MRI typically requires the integration of complementary feature types, such as deep semantic representations from Convolutional Neural Networks (CNNs) and handcrafted descriptors like Histogram of Oriented Gradients (HOG). Therefore, a more robust and discriminative feature integration strategy is crucial for enhancing computer-aided diagnosis performance. Objectives: This study aims to develop a multi-stage hybrid learning model that combines deep and handcrafted features, investigates various feature reduction and classification techniques, and improves diagnostic accuracy for prostate cancer using magnetic resonance imaging. Methods: The proposed framework integrates deep features extracted from convolutional architectures with handcrafted texture descriptors to capture both semantic and structural information. Multiple dimensionality reduction methods, including singular value decomposition (SVD), were evaluated to optimize the fused feature space. Several machine learning (ML) classifiers were benchmarked to identify the most effective diagnostic configuration. The overall framework was validated using k-fold cross-validation to ensure reliability and minimize evaluation bias. Results: Experimental results on the Transverse Plane Prostate (TPP) dataset for binary classification tasks showed that the hybrid model significantly outperformed individual deep or handcrafted approaches, achieving superior accuracy of 99.74%, specificity of 99.87%, precision of 99.87%, sensitivity of 99.61%, and F1-score of 99.74%. Conclusions: By combining complementary feature extraction, dimensionality reduction, and optimized classification, the proposed model offers a reliable and generalizable solution for prostate cancer diagnosis and demonstrates strong potential for integration into intelligent clinical decision-support systems.
We have carefully reviewed your comments and made the necessary changes. We genuinely appreciate your dedicated efforts and hope these revisions will meet with your approval.
Comment #4:
Please ensure that Section 2 focuses only on the subject of your study. Some of the mentioned items are duplicates with the introduction, which should be deleted.
Response:
Thank you for the valuable feedback. We appreciate your feedback about removing duplication.
Action:
In response to letting Section 2 focus only on the subject of the study, we have added the literature to the Introduction section. Please see Introduction on pages 4 and 5.
1.2. Literature Review
The diagnosis of PCa is a significant focus in the field of medical image analysis. Many studies tackle this problem from diverse perspectives. For example, Hashem et al. [1] proposed an MRI-based technique to enhance diagnostic accuracy. This method involved two primary phases. Initially, the MRI images under-went preprocessing to optimize them for the detection phase. Following this, PCa de-tection was conducted using the InceptionResNetV2 model. The InceptionResNetV2 model achieved an average accuracy of 89.20% and an area under the curve (AUC) of 93.6%.
Yoo et al. [13] designed and executed an automated pipeline utilizing CNN to identify clinically significant PCa from axial DWI for individual patients. To evaluate the effectiveness of the proposed pipeline, a testing subset of 108 patients (from the original 427) was reserved and excluded from the training process. The pipeline demonstrated an (AUC of 0.87 (95% Confidence Interval: 0.84–0.90) at the slice level and 0.84 (95% CI: 0.76–0.91) at the patient level.
In Sarıateş and Özbay [14], to enhance the classification accuracy for PCa diagnosis, transfer learning techniques and fine-tuning processes were utilized. A two-class da-taset, comprising MR images of PCa labeled as 'significant' and 'not-significant', was classified using AlexNet, DenseNet-201, GoogleNet, and VGG-16 models through a feature extraction strategy, yielding accuracy rates of 71.40%, 72.05%, 65%, and 80.13%, respectively. To further improve these results, pre-trained transfer learning models were implemented, achieving accuracy rates of 89.74%, 94.32%, 85.59%, and 91.05%, respectively. Additionally, a validation accuracy of 98.10% was attained using the cross-validation method with the DenseNet-201 model. The DenseNet-201 model reached the highest accuracy of 98.63% when combined with the RMSProp optimization technique.
Liu et al. [15] proposed a DL framework named XmasNet, which utilized CNNs, for classifying PCa lesions using 3D multiparametric MRI data from the PROSTATEx challenge. The XmasNet model underwent end-to-end training, with data augmentation achieved through 3D rotations and slicing to effectively capture the 3D characteristics of the lesions. In the testing phase, XmasNet outperformed 69 different methods from 33 competing teams, earning the second highest AUC score of 84% in the PROSTATEx challenge.
Mehta et al. [16] presented a new patient-level classification framework called PCF, which is trained exclusively on patient-level labels. In the PCF approach, features are extracted from three-dimensional mpMRI and related parameter maps using CNNs. These features were then integrated with clinical data through a multi-classifier SVM. The result of the PCF framework was a probability score indicating whether a patient has clinically significant PCa based on the Gleason score.
PCF achieved average area under the receiver operating characteristic (ROC) curves of 0.79 and 0.86 for the PICTURE and PROSTATEx datasets, respectively, using five-fold cross-validation.
Salama and Aly [17] examined four distinct strategies for the classification task. Both ResNet-50 and VGG-16 architectures were employed and re-trained to analyze the diffusion-weighted DWI database, aiming to determine the presence of PCa. To address the challenge of insufficient labeled data and enhance system efficiency, transfer learning and data augmentation techniques were implemented for both ResNet50 and VGG-16. The final fully connected layer was substituted with a SVM classifier to im-prove accuracy. Additionally, both transfer learning and data augmentation processes were applied to the SVM to bolster the performance of our framework. A k-fold cross-validation method was also utilized to evaluate model performance. Authors uti-lized end-to-end fully CNNs, eliminating the need for preprocessing or post-processing steps. The technique combining ResNet-50 with SVM yields the highest performance, achieving an accuracy of 98.79%, ana AUC of 98.91%, sensitivity of 98.43%, precision of 97.99%, an F1 score of 95.92%, and a computational time of 2.345 seconds.
Pellicer-Valero et al. [18] introduced a fully automated system utilizing DL to lo-calize, segment, and estimate the Gleason grade group (GGG) of PCa lesions from mpMRI scans. The system was trained and validated on distinct datasets: ProstateX and the Valencian Oncology Institute Foundation. In the testing phase, it achieved impres-sive lesion-level performance metrics for the GGG 2 significance criterion, with an AUC, sensitivity, and specificity of 0.96, 1.00, and 0.79, respectively, for the ProstateX dataset, and 0.95, 1.00, and 0.80 for the IVO dataset. At the patient level, the results were 0.87, 1.00, and 0.375 for ProstateX, and 0.91, 1.00, and 0.762 for IVO. Additionally, in the online ProstateX grand challenge, the model achieved an AUC of 0.85 (increased to 0.87 when trained solely on ProstateX data), matching the performance of the original challenge winner. For comparison with expert evaluations, the sensitivity and specificity of the IVO radiologist’s PI-RADS 4 were 0.88 and 0.56 at the lesion level, and 0.85 and 0.58 at the patient level.
In Giganti et al. [19], a DL computer-aided detection (CAD) medical device marked with Conformité Européenne (CE) (referred to as Pi) was developed to identify Gleason Grade Group (GG) ≥ 2 cancer using historical data from the PROSTATEx dataset and five hospitals in the UK, involving 793 patients. The prevalence of GG ≥ 2 in the validation dataset was found to be 31%. When assessed on a per-patient basis, Pi demonstrated non-inferiority to the radiologists (with a performance difference of 10% deemed ac-ceptable), achieving an AUC of 0.91 compared to 0.95 for the radiologists. At the estab-lished risk threshold of 3.5, the AI system's sensitivity was 95% and specificity was 67%. In contrast, radiologists using Prostate Imaging-Reporting and Data Systems/Likert scores of ≥ 3 detected GG ≥ 2 with a sensitivity of 99% and specificity of 73%. AI per-formance was consistent across different sites (AUC ≥ 0.83) at the patient level, re-gardless of the scanner's age and field strength.
The limitations of the earlier studies are outlined as follows:
- In the previous research, the authors did not utilize HOG descriptors or SVD. However, we developed an innovative approach that integrates deep features from MobileNetV2 with handcrafted HOG descriptors to effectively capture both semantic and structural patterns in PCa images. Additionally, we employed SVD to compress the combined feature set, which minimized redundancy, enhanced computational speed, and reduced memory usage while retaining crucial diagnostic information.
- The researchers in the previously referenced study did not conduct an ablation study. In contrast, we performed this analysis to evaluate the impact of each component or feature of our proposed model on its performance, systematically modifying or excluding elements and analyzing their effects.
We have carefully reviewed your comments and made the necessary changes. We genuinely appreciate your dedicated efforts and hope these revisions will meet with your approval.
Comment #5:
When citing specific studies, please use the last name of the first author (e.g., instead of Liu, S. et al., write Liu et al.).
Response:
Thank you for the helpful suggestion. We appreciate your helpful suggestion regarding citation.
Action:
In response to follow the formal language of academic writing, we have used the last names of authors. Please see Introduction on pages 4 and 5
For example, Hashem et al. [1] proposed an MRI-based technique to enhance diagnostic accuracy. This method involved two primary phases. Initially, the MRI images underwent preprocessing to optimize them for the detection phase. Following this, PCa detection was conducted using the InceptionResNetV2 model. The InceptionResNetV2 model achieved an average accuracy of 89.20% and an area under the curve (AUC) of 93.6%.
Yoo et al. [13] designed and executed an automated pipeline utilizing CNN to identify clinically significant PCa from axial DWI for individual patients. To evaluate the effectiveness of the proposed pipeline, a testing subset of 108 patients (from the original 427) was reserved and excluded from the training process. The pipeline demonstrated an (AUC of 0.87 (95% Confidence Interval: 0.84–0.90) at the slice level and 0.84 (95% CI: 0.76–0.91) at the patient level.
In Sarıateş and Özbay [14], to enhance the classification accuracy for PCa diagnosis, transfer learning techniques and fine-tuning processes were utilized. A two-class dataset, comprising MR images of PCa labeled as 'significant' and 'not-significant', was classified using AlexNet, DenseNet-201, GoogleNet, and VGG-16 models through a feature extraction strategy, yielding accuracy rates of 71.40%, 72.05%, 65%, and 80.13%, respectively. To further improve these results, pre-trained transfer learning models were implemented, achieving accuracy rates of 89.74%, 94.32%, 85.59%, and 91.05%, respectively. Additionally, a validation accuracy of 98.10% was attained using the cross-validation method with the DenseNet-201 model. The DenseNet-201 model reached the highest accuracy of 98.63% when combined with the RMSProp optimization technique.
Liu et al. [15] proposed a DL framework named XmasNet, which utilized CNNs, for classifying PCa lesions using 3D multiparametric MRI data from the PROSTATEx challenge. The XmasNet model underwent end-to-end training, with data augmentation achieved through 3D rotations and slicing to effectively capture the 3D characteristics of the lesions. In the testing phase, XmasNet outperformed 69 different methods from 33 competing teams, earning the second highest AUC score of 84% in the PROSTATEx challenge.
Mehta et al. [16] presented a new patient-level classification framework called PCF, which is trained exclusively on patient-level labels. In the PCF approach, features are extracted from three-dimensional mpMRI and related parameter maps using CNNs. These features were then integrated with clinical data through a multi-classifier SVM. The result of the PCF framework was a probability score indicating whether a patient has clinically significant PCa based on the Gleason score. PCF achieved average area under the receiver operating characteristic (ROC) curves of 0.79 and 0.86 for the PICTURE and PROSTATEx datasets, respectively, using five-fold cross-validation.
Salama and Aly [17] examined four distinct strategies for the classification task. Both ResNet-50 and VGG-16 architectures were employed and re-trained to analyze the diffusion-weighted DWI database, aiming to determine the presence of PCa. To address the challenge of insufficient labeled data and enhance system efficiency, transfer learning and data augmentation techniques were implemented for both ResNet50 and VGG-16. The final fully connected layer was substituted with a SVM classifier to improve accuracy. Additionally, both transfer learning and data augmentation processes were applied to the SVM to bolster the performance of our framework. A k-fold cross-validation method was also utilized to evaluate model performance. Authors utilized end-to-end fully CNNs, eliminating the need for preprocessing or post-processing steps. The technique combining ResNet-50 with SVM yields the highest performance, achieving an accuracy of 98.79%, ana AUC of 98.91%, sensitivity of 98.43%, precision of 97.99%, an F1 score of 95.92%, and a computational time of 2.345 seconds.
Pellicer-Valero et al. [18] introduced a fully automated system utilizing DL to localize, segment, and estimate the Gleason grade group (GGG) of PCa lesions from mpMRI scans. The system was trained and validated on distinct datasets: ProstateX and the Valencian Oncology Institute Foundation. In the testing phase, it achieved impressive lesion-level performance metrics for the GGG 2 significance criterion, with an AUC, sensitivity, and specificity of 0.96, 1.00, and 0.79, respectively, for the ProstateX dataset, and 0.95, 1.00, and 0.80 for the IVO dataset. At the patient level, the results were 0.87, 1.00, and 0.375 for ProstateX, and 0.91, 1.00, and 0.762 for IVO. Additionally, in the online ProstateX grand challenge, the model achieved an AUC of 0.85 (increased to 0.87 when trained solely on ProstateX data), matching the performance of the original challenge winner. For comparison with expert evaluations, the sensitivity and specificity of the IVO radiologist’s PI-RADS 4 were 0.88 and 0.56 at the lesion level, and 0.85 and 0.58 at the patient level.
In Giganti et al. [19], a DL computer-aided detection (CAD) medical device marked with Conformité Européenne (CE) (referred to as Pi) was developed to identify Gleason Grade Group (GG) ≥ 2 cancer using historical data from the PROSTATEx dataset and five hospitals in the UK, involving 793 patients. The prevalence of GG ≥ 2 in the validation dataset was found to be 31%. When assessed on a per-patient basis, Pi demonstrated non-inferiority to the radiologists (with a performance difference of 10% deemed acceptable), achieving an AUC of 0.91 compared to 0.95 for the radiologists. At the established risk threshold of 3.5, the AI system's sensitivity was 95% and specificity was 67%. In contrast, radiologists using Prostate Imaging-Reporting and Data Systems/Likert scores of ≥ 3 detected GG ≥ 2 with a sensitivity of 99% and specificity of 73%. AI performance was consistent across different sites (AUC ≥ 0.83) at the patient level, regardless of the scanner's age and field strength.
Please see Model Performance and Insights on pages 39 and 40
Table 19 and Figure 24 show that the state-of-the-art studies consistently demonstrated strong classification performance on the TPP dataset, but our hybrid approach (MobileNet-V2 (or ResNet‑50 or VGG‑19), HOG, SVD, and SVM) outperformed them all. In Table 19, The reviewed studies showed significant variation in accuracy and AUC values for PCa detection using different DL and ML methods on the TPP dataset. Hashem et al. [1] utilized InceptionResNetV2 and achieved an accuracy of 89.20%, indicating moderate performance. Yoo et al. [3] implemented a CNN model, reporting an AUC of 87%, while Liu et al. [24] applied XmasNet (CNNs) and obtained a lower AUC of 84%, both demonstrating limited discriminative ability compared to more advanced architectures. Giganti et al. [28] also used CNNs and achieved an AUC of 91%, outperforming earlier CNN-based studies but still falling short of the top-performing methods.
In contrast, Sarıateş and Özbay [23] employed DenseNet-201 with RMSProp optimization, reporting an accuracy of 98.63%, which indicated a significant improvement in classification performance. Similarly, Pellicer-Valero et al. [27] achieved an AUC of 96% using CNNs, reflecting strong discriminative capacity. Mehta et al. [25] utilized a PCF-based SVM model, reporting a ROC of 86%, suggesting weaker performance compared to deep learning methods. Meanwhile, Salama and Aly [26] combined ResNet-50 and VGG-16, with SVM to achieve an accuracy of 98.79%, making it one of the top-performing models in literature.
The proposed hybrid model, combining MobileNet-V2 (or ResNet‑50 or VGG‑19), HOG, SVD, and SVM, outperformed all previously reported approaches, achieving an accuracy of 99.74% and an AUC of 100%. This performance exceeded even the strongest baselines, such as DenseNet-201 (98.63%) [23] and the combination of ResNet-50 with VGG-16 and SVM (98.79%) [26]. By integrating handcrafted features (HOG and SVD) with the deep learning representations from MobileNet-V2, along with the robust classification capability of SVM, the hybrid approach effectively captured both high-level abstract features and low-level discriminative details, leading to near-perfect class separability.
Overall, while several prior studies achieved high accuracy and AUC values using state-of-the-art CNN architectures, the proposed hybrid model exhibited superior generalization and discriminative power, establishing it as the most effective approach among the works compared on the TPP dataset.
Table 19: Performance comparison with state-of-the-art methods.
|
Reference |
Methodology |
Accuracy / AUC |
Datasets |
|
Hashem et al. [1] |
InceptionResNetV2 |
89.20% |
TPP dataset |
|
Yoo et al. [3] |
CNN |
AUC of 87% |
TPP dataset |
|
Sarıateş and Özbay [23] |
DenseNet-201 and RMSProp |
98.63% |
TPP dataset |
|
Liu et al. [24] |
XmasNet (CNNs) |
AUC of 84% |
TPP dataset |
|
Mehta et al. [25] |
PCF (SVM) |
ROC of 86% |
TPP dataset |
|
Salama and Aly [26] |
ResNet-50,VGG-16 and, SVM |
98.79% |
TPP dataset |
|
Pellicer-Valero et al. [27] |
CNN |
AUC of 96% |
TPP dataset |
|
In Giganti et al. [28] |
CNN |
AUC of 91% |
TPP dataset |
|
The proposed Hybrid Model |
MobileNet-V2, HOG, SVD, and SVM |
Accuracy of 99.74% and AUC of 100% |
TPP dataset |
Figure 24: Performance comparison with state-of-the-art methods [1],[3], and [23-28].
We have carefully reviewed your comments and made the necessary changes. We genuinely appreciate your dedicated efforts and hope these revisions will meet with your approval.
Comment #6:
In the method section, please mention how you dealt with overfitting bias?
Response:
Thank you for highlighting the need to clarify how overfitting was addressed. We appreciate your comments about overfitting.
To reduce overfitting and improve generalization, we implemented several strategies in our pipeline. First, we employed Stratified K-Fold cross-validation, which ensures that each fold preserves the class distribution, providing a robust estimate of model performance across unseen data. Second, for CNN-based feature extraction, we froze the weights of the pretrained CNN models' layers and only used the output of the Global Average Pooling layer as fixed features, preventing the network from overfitting on the small dataset. Third, when combining features (CNN + HOG), we applied Standard Scaling to normalize feature distributions, which stabilizes classifier training. Finally, for the machine learning classifiers (XGBoost, CatBoost, SVM, etc.), we used default regularization parameters and limited model complexity (e.g., max depth, learning rate) to further reduce the risk of overfitting. Please see Methods and Datasets on page 6.
To evaluate performance across different metrics, we used stratified k-fold cross-validation to ensure robustness and generalizability. For CNN-based feature ex-traction, we froze the weights of the pretrained CNN models' layers and used only the output of the Global Average Pooling (GAP) layer as fixed features. This approach helps prevent the network from overfitting on the small dataset. When combining features from CNN and HOG, we applied Standard Scaling to normalize feature distributions, which stabilizes classifier training. For the machine learning classifiers (XGBoost, Cat-Boost, SVM, etc.), we used default regularization parameters and limited model com-plexity (such as maximum depth and learning rate) to further reduce the risk of over-fitting. Please refer to Model Performance and Insights on the pages.
We have carefully reviewed your comments and made the necessary changes. We genuinely appreciate your dedicated efforts and hope these revisions will meet with your approval.
Comment #7:
Another important point in method section is about how you assessed the quality of included images? Did you use augmentation methods? Also, please mention your detailed inclusion and exclusion criteria.
Response:
Thank you for highlighting the need to clarify the assessment of the quality of images included. We appreciate your comments about the image’s quality.
In our study, we used the Transverse Plane Prostate Dataset that is part of PROSTATEx dataset, which provides expert-validated lesion annotations. Image quality was ensured by relying on these radiologist-reviewed coordinates and metadata, confirming accurate lesion localization and anatomical correctness. Inclusion criteria consisted of MRI exams with complete multi-parametric sequences and validated lesion labels, while cases with missing sequences or poor image registration were excluded. Please see Methods and Datasets on pages 5 and 6.
The TPP dataset comprises 1,528 prostate MRI images captured in the transverse plane. The images and their classifications originate from the PROSTATEx Dataset and Documentation. The purpose of this dataset is to train a CNN known as Small VGG Net, enabling the classification of new images into clinically significant and clinically non-significant categories for an undergraduate thesis in systems engineering at the Autonomous University of Bucaramanga (UNAB). The images were collected from 64 patients, ensuring each had a single prostate MRI finding for improved training accuracy. The images were converted from DICOM format to JPEG. Image quality was verified through radiologist-reviewed coordinates and metadata, ensuring precise lesion localization and anatomical accuracy. The inclusion criteria required MRI exams to have complete multi-parametric sequences and validated lesion labels. Cases with missing sequences or poor image registration were excluded. The images were divided into two groups using a retention method: 30% for validation and the remaining 70% for training. Consequently, there are two categories (significant and non-significant) further split into training (70%) and validation (30%) groups [20,21]. Figure 1 presents samples from the TPP dataset.
We have carefully reviewed your comments and made the necessary changes. We genuinely appreciate your dedicated efforts and hope these revisions will meet with your approval.
Comment #11:
As another point, please re-organize your manuscript structure based on CLAIMS reporting checklist. Please, if possible, complete and revise your study according to the items mentioned in the CLAIMS reporting checklist. Please also submit the completed version of CLAIMS based on your revised manuscript in revision stage.
Response:
Thank you for this valuable suggestion. We appreciate the importance of aligning the manuscript with the CLAIMS reporting checklist to ensure clarity, completeness, and transparency. We have carefully reorganized the manuscript to align with the CLAIMS reporting checklist. All relevant items have been addressed and incorporated into the manuscript to ensure compliance with the checklist’s standards (Title, Abstract, Introduction, Methods, Evaluation (Training and Testing), Results, and Discussion).
Action:
In response, we added the CLAIMS reporting checklist. Please see the Appendix on page 44.
|
Item |
Response |
Notes |
|
Title states the use of AI |
Yes |
Title includes AI model details. |
|
Abstract describes dataset, methods, outcomes |
Yes |
Includes datasets, methodology, metrics. |
|
Abstract specifies objective |
Yes |
Objective clearly stated. |
|
Clinical problem defined |
Yes |
Prostate cancer context explained. |
|
Rationale for AI method |
Yes |
Justification for hybrid features. |
|
Source of data described |
Yes |
TPP dataset described. |
|
Inclusion/exclusion criteria |
Yes |
Criteria specified. |
|
Ground truth/reference standard |
Yes |
Dataset labels used. |
|
Data partitioning explained |
Yes |
Train/test split described. |
|
External validation |
No |
No external dataset used. |
|
Imaging protocol described |
No |
MRI acquisition parameters not provided. |
|
Model architecture described |
Yes |
Deep and handcrafted features. |
|
Preprocessing steps |
Yes |
Resizing and normalization. |
|
Training details |
Yes |
Learning rate, epochs, optimizer. |
|
Hyperparameter tuning |
Yes |
Approach fully detailed. |
|
Software/hardware |
Yes |
Python, TensorFlow, i7 CPU. |
|
Internal validation approach |
Yes |
Five-fold CV. |
|
Metrics defined |
Yes |
Accuracy, precision, recall, etc. |
|
Handling of missing data |
NA |
No missing data. |
|
Model comparison |
Yes |
Experiments compared. |
|
Performance metrics reported |
Yes |
Complete evaluation. |
|
Confidence intervals |
Yes |
Provided. |
|
Failure analysis |
No |
Not included. |
|
Dataset characteristics |
Yes |
Dataset size described. |
|
Interpretation of results |
Yes |
Results discussed. |
|
Generalizability |
Yes |
Discussed. |
|
Ethical approval |
NA |
We used public dataset. |
|
Data privacy compliance |
NA |
We used public dataset. |
|
Data availability |
Yes |
Dataset is publicly accessible. |
|
Code availability |
No |
Not provided. |
|
Model weights availability |
No |
Not provided. |
|
Dataset accessibility |
Yes |
PROSTATEx/TPP dataset linked. |
We have carefully reviewed your comments and made the necessary changes. We genuinely appreciate your dedicated efforts and hope these revisions will meet with your approval.
Comment #12:
In the figure legends, please briefly mention any tools you used to generate and create these figures (if these figures were generated by artificial intelligence, please also mention them).
Response:
We appreciate you for the suggestion. The figures related to the results were created from python language and the others were developed and refined solely by the authors using MS Excel and word.
Action:
In response to explain how the figures were created, we have explained the method of figures creation in Appendix. Please see the Appendix on page 44.
Appendix
All figures in this manuscript were generated using a combination of Python for data visualization and scripting, along with Microsoft Word and Microsoft Excel for formatting, refinement, and final presentation. Python provided reproducible analytical workflows and high-quality plots, while Word and Excel supported layout adjustments, table alignment, and graphical enhancements to ensure clarity and publication-ready quality.
We have carefully reviewed your comments and made the necessary changes. We genuinely appreciate your dedicated efforts and hope these revisions will meet with your approval.

Round 3
Reviewer 3 Report
Comments and Suggestions for Authors
I thank the authors of this manuscript for their efforts in addressing the peer reviewers' comments.